# Guarantees for Self-Play in Multiplayer Games via Polymatrix Decomposability

**Revan MacQueen**
Department of Computing Science
University of Alberta / Amii

revan@ualberta.ca

**James R. Wright**
Department of Computing Science
University of Alberta / Amii

james.wright@ualberta.ca

## Abstract

Self-play is a technique for machine learning in multi-agent systems where a learning algorithm learns by interacting with copies of itself. Self-play is useful for generating large quantities of data for learning, but has the drawback that the agents the learner will face post-training may have dramatically different behavior than the learner came to expect by interacting with itself. For the special case of two-player constant-sum games, self-play that reaches Nash equilibrium is guaranteed to produce strategies that perform well against any post-training opponent; however, no such guarantee exists for multiplayer games. We show that in games that approximately decompose into a set of two-player constant-sum games (called constant-sum polymatrix games) where global $\epsilon$-Nash equilibria are boundedly far from Nash equilibria in each subgame (called subgame stability), any no-external-regret algorithm that learns by self-play will produce a strategy with bounded vulnerability. For the first time, our results identify a structural property of multiplayer games that enable performance guarantees for the strategies produced by a broad class of self-play algorithms. We demonstrate our findings through experiments on Leduc poker.

## 1 Introduction

Self-play is one of the most commonly used approaches for machine learning in multi-agent systems. In self-play, a learner interacts with copies of itself to produce data that will be used for training. Some of the most noteworthy successes of AI in the past decade have been based on self-play; by employing the procedure, algorithms have been able to achieve super-human abilities in various games, including Poker (Moravcik et al., 2017; Brown & Sandholm, 2018, 2019), Go and Chess (Silver et al., 2016, 2018), Starcraft (Vinyals et al., 2019), Diplomacy (Paquette et al., 2019), and Stratego (Perolat et al., 2022).

Self-play has the desirable property that unbounded quantities of training data can be generated (assuming access to a simulator). But using self-play necessarily involves a choice of agents for the learner to train with: copies of itself. Strategies that perform well during training may perform poorly against new agents, whose behavior may differ dramatically from that of the agents that the learner trained against.

The problem of learning strategies during training that perform well against new agents is a central challenge in algorithmic game theory and multi-agent reinforcement learning (MARL) (Matignon et al., 2012; Lanctot et al., 2017). In particular, a self-play trained agent interacting with agents from an independent self-play instance—differing only by random seed—can lead to dramatically worse performance (Lanctot et al., 2017). The self-play-based DORA (Bakhtin et al., 2021) performs well aginst copies of itself in the game of no-press Diplomacy, but poorly against human-like agents

37th Conference on Neural Information Processing Systems (NeurIPS 2023).

(Bakhtin et al., 2022). Self-play has also been known to perform poorly against new agents in games with highly specialized conventions (Hu et al., 2020), such as Hanabi (Bard et al., 2020). In training, one instance of a self-play algorithm may learn conventions that are incompatible with the conventions of another instance.

There are special classes of games where the strategies learned through self-play generalize well to new agents. In two-player, constant-sum games there exist strong theoretical results guaranteeing the performance of a strategy learned through self-play: Nash equilibrium strategies are maxmin strategies, which will perform equally well against any optimal opponent and can guarantee the value of the game against any opponent.

We lose these guarantees outside of two-player constant-sum games. For example, consider the simple two-player coordination game of Figure 1. If both players choose the same action, both receive a utility of 1, otherwise they receive 0. Suppose the row player learned in self-play to choose $a$ (which performs well against another $a$-player). Similarly, column learned to play $b$. If these two players

|       | $a$      | $b$      |
|-------|----------|----------|
| $a$   | $1,1$    | $0,0$    |
| $b$   | $0,0$    | $1,1$    |

Figure 1: A simple coordination game

played against each other, both agents would regret their actions. Upon the introduction of a new agent who did not train with a learner, despite $a$ and $b$ being optimal strategies during training, they fail to generalize to new agents. As this example demonstrates, equilibrium strategies in general are *vulnerable*: agents are not guaranteed the equilibrium's value against new agents–even if the new agent's also play an equilibrium strategy.

In multiplayer, general-sum games, no-regret self-play is no longer guaranteed to produce Nash equilibra—instead, algorithms converge to a *mediated equilibrium*, where a mediator recommends actions to each player (Von Stengel & Forges, 2008; Farina et al., 2019, 2020; Morrill et al., 2021b). The mediator can represent an external entity that makes explicit recommendations, such as traffic lights mediating traffic flows. More commonly in machine learning, correlation can arise through the shared history of actions of learning agents interacting with each other (Hart & Mas-Colell, 2000). If a strategy learned in self-play were played against new agents, these new agents may not have access to the actions taken by other agents during training, so agents would no longer be able to correlate their actions. In fact, even if all agents play a decorrelated strategy from *the same* mediated equilibrium, the result may not be an equilibrium.[1]

Despite the problems of vulnerability and loss of correlation, self-play has shown promising results outside of two-player constant-sum games. For example, algorithms based on self-play have outperformed professional poker players in multiplayer Texas hold 'em, despite the lack of theoretical guarantees (Brown & Sandholm, 2019).

We seek to understand what structure in multiplayer games will allow self-play to compute a good strategy. We show that any multiplayer game can be projected into the set of two-player constant-sum games between each pair of players, called *constant-sum polymatrix games*. The closer a game is to this space, the less the problem of correlation affects the removal of a mediator. We identify a second important property, called *subgame stability*, where global $\epsilon$-Nash equilibria are boundedly far from Nash equilibria in each two-player subgame. We show that if a multiplayer, general-sum game is close to a subgame stable constant-sum polymatrix game, this is sufficient for strategies learned via self-play to generalize to new agents, as they do in two-player constant-sum games.

Throughout this work, we take an algorithm-agnostic approach by assuming only that self-play is performed by a *regret minimizing algorithm*. This is accomplished by analyzing directly the equilibria that no-regret algorithms converge to—namely coarse correlated equilibria. As a result, our analysis applies to a broad class of game-theoretically-inspired learning algorithms but also to MARL algorithms that converge to coarse correlated equilibria (Marris et al., 2021; Liu et al., 2021; Jin et al., 2021), since any policy can be transformed into a mixed strategy with Kuhn's Theorem (Kuhn, 1953). For the remainder of this work, when we say "self-play" we are referring to self-play using a no-regret algorithm.

Decomposition-based approaches have been used in prior work to show convergence of fictitious play to Nash equilibria in two-player games (Chen et al., 2022) and evolutionary dynamics (Tuyls et al., 2018). Cheung & Tao (2020) decompose games into zero-sum and cooperative parts to analyse

---

[1]Please refer to the Appendix A for an example

the chaotic nature of Multiplicative Weights and Follow-the-Regularized-Leader. We focus less on the convergence of algorithms per se, and focus instead on the generalization of learned strategies to new agents. We are, to the best of our knowledge, the first to do so.

After defining our structural properties and proving our main results, we conclude with experiments on Leduc poker to elucidate why self-play performs well in multiplayer poker. Our results suggest that regret-minimization techniques converge to a subset of the game's strategy space that is well-approximated by a subgame stable constant-sum polymatrix game.

## 2  Background

**Normal Form Games**  A normal form game is a 3 tuple $G = (N, \mathrm{P}, u)$ where $N$ is a set of players, $\mathrm{P} = \bigtimes_{i \in N} \mathrm{P}_i$ is a joint pure strategy space and $\mathrm{P}_i$ is a set of *pure strategies* for player $i$. Let $n = |N|$. Pure strategies are deterministic choices of actions in the game. We call $\rho \in \mathrm{P}$ a *pure strategy profile*. $u = (u_i)_{i \in N}$ is a set of *utility functions* where $u_i : \mathrm{P} \to \mathbb{R}$. A player $i$ can randomize by playing a *mixed strategy*, a probability distribution $s_i$ over $i$'s pure strategies. Let $S_i = \Delta(\mathrm{P}_i)$ be the set of player $i$'s mixed strategies (where $\Delta(X)$ denotes the set of probability distributions over a domain $X$), and let $S = \bigtimes_{i \in N} S_i$ be the set of mixed strategy profiles. We overload the definition of utility function to accept mixed strategies as follows: $u_i(s) = \sum_{\rho \in \mathrm{P}} \left( \prod_{i \in N} s_i(\rho_i) \right) u_i(\rho)$. We use $\rho_{-i}$ and $s_{-i}$ to denote a joint assignment of pure (resp. mixed) strategies to all players except for $i$, thus $s = (s_i, s_{-i})$. We use $\mathrm{P}_{-i}$ and $S_{-i}$ to denote the sets of all such assignments.

**Hindsight Rationality**  The hindsight rationality framework (Morrill et al., 2021b) conceptualizes the goal of an agent as finding a strategy that minimizes regret with respect to a set of deviations $\Phi$. A deviation $\phi \in \Phi$ is a mapping $\phi : S_i \to S_i$ that transforms a learner's strategy into some other strategy. Regret measures the amount the learner would prefer to deviate to $\phi(s_i)$: $u_i(\phi(s_i), s_{-i}) - u_i(s_i, s_{-i})$. An agent is hindsight rational with respect to a set of deviations $\Phi$ if the agent does not have positive regret with respect to any deviation in $\Phi$, i.e. $\forall \phi \in \Phi, u_i(\phi(s_i), s_{-i}) - u_i(s_i, s_{-i}) \leq 0$. Let $\mu \in \Delta(\mathrm{P})$ be a distribution over pure strategy profiles and $(\Phi_i)_{i \in N}$ be a choice of deviation sets for each player.

**Definition 2.1** ($\epsilon$-Mediated Equilibrium (Morrill et al., 2021b)).  We say $m = (\mu, (\Phi_i)_{i \in N})$ is an *$\epsilon$-mediated equilibrium* if $\forall i \in N, \phi \in \Phi_i$ we have $\mathbb{E}_{\rho \sim \mu} [u_i(\phi(\rho_i), \rho_{-i}) - u_i(\rho)] \leq \epsilon$. A *mediated equilibrium* is a 0-mediated equilibrium.

Learning takes place in an online learning environment. At each iteration $t$, a learning agent $i$ chooses a strategy $s_i^t$ while all other agents choose a strategy profile $s_{-i}^t$. No-$\Phi$-regret learning algorithms ensure that the maximum average positive regret tends to 0:

$$\lim_{T \to \infty} \frac{1}{T} \left( \max_{\phi \in \Phi} \sum_{t=1}^{T} u_i(\phi(s_i^t), s_{-i}^t) - u_i(s_i^t, s_{-i}^t) \right) \to 0.$$

If all agents use a no-regret learning algorithms w.r.t. a set of deviations $\Phi_i$, the *empirical distribution of play* converges to a mediated equilibrium. Formally, let $\hat{\mu} \in \Delta(\mathrm{P})$ be the empirical distribution of play, where the weight on $\rho \in \mathrm{P}$ is $\hat{\mu}(\rho) \doteq \sum_{t=1}^{T} \left( \prod_{i \in N} s_i^t(\rho_i) \right)$. As $T \to \infty$, $\hat{\mu}$ converges to $\mu$ of a mediated equilibrium $(\mu, (\Phi_i)_{i \in N})$.

Different sets of deviations determine the strength of a mediated equilibrium. For normal-form games, the set of *swap* deviations, $\Phi_{SW}$, are all possible mappings $\phi : \mathrm{P}_i \to \mathrm{P}_i$. We may apply a swap deviation $\phi$ to a mixed strategy $s_i$ by taking its pushforward measure: $[\phi(s_i)](\rho_i) = \sum_{\rho_i' \in \phi^{-1}(\rho_i)} s_i(\rho_i')$, where $\phi^{-1}(\rho_i) = \{\rho_i' \in \mathrm{P}_i \mid \phi(\rho_i') = \rho_i\}$. The set of *internal* deviations $\Phi_I$, which replace a single pure strategy with another, offer the same strategic power as swap deviations (Foster & Vohra, 1999). Formally, $\Phi_I = \{\phi \in \Phi_{SW} \mid \exists \rho_i, \rho_i' : [\phi(\rho_i) = \rho_i'] \wedge [\forall \rho_i'' \neq \rho_i, \phi(\rho_i'') = \rho_i'']\}$. The set of *external* deviations, $\Phi_{EX}$, is even more restricted: any $\phi \in \Phi_{EX}$ maps all (mixed) strategies to some particular pure strategy; i.e. $\Phi_{EX} = \{\phi \in \Phi_{SW} \mid \exists \rho_i : \forall \rho_i', \phi(\rho_i') = \rho_i\}$. The choice of $(\Phi_i)_{i \in N}$ determines the nature of the mediated equilibrium—provided the learning algorithm for player $i$ is no-$\Phi_i$-regret (Greenwald et al., 2011). For example, if all players are hindsight rational w.r.t. $\Phi_{EX}$, then $\hat{\mu}$ converges to the set of coarse correlated equilibria (CCE) (Moulin & Vial, 1978) and if all players are hindsight rational w.r.t. $\Phi_I$ then $\hat{\mu}$ converges to the set of correlated equilibria (Aumann, 1974).

A special case of mediated equilibria are *Nash equilibria*. If some mediated equilibrium $m = (\mu, (\Phi_i)_{i \in N})$ is a product distribution (i.e. $\mu(\rho) = \prod_{i \in N} s_i(\rho) \; \forall \rho \in P$ ) and $\Phi_i \supseteq \Phi_{EX} \; \forall i \in N$ then $\mu$ is a Nash equilibrium. Similarly, an $\epsilon$-mediated equilibrium is an $\epsilon$-Nash equilibrium if $\mu$ is a product distribution and $\Phi_i \supseteq \Phi_{EX} \; \forall i \in N$.

In sequential decision making scenarios (often modelled as extensive form games), the set of deviations is even more rich (Morrill et al., 2021b). All of these deviation classes—with the exception of action deviations (Selten, 1988) (which are so weak they do not even imply Nash equilibria in two-player constant-sum games, see appendix)—are stronger than external deviations. This means that the equilibria of any algorithm that minimizes regret w.r.t. a stronger class of deviations than external deviations still inherit all the properties of CCE (for example Hart & Mas-Colell (2000); Zinkevich et al. (2008); Celli et al. (2020); Steinberger et al. (2020); Morrill et al. (2021a)). Thus, we focus on CCE since the analysis generalizes broadly. Moreover, CCE can be computed efficiently, either analytically (Jiang & Leyton-Brown, 2011) or by a learning algorithm. When we refer to CCE, we use the distribution $\mu$ to refer to the CCE, since $\Phi$ is implicit.

## 3   Vulnerability

The choice of other agents during learning affects the strategy that is learned. Choosing which agents make good "opponents" during training is an open research question (Lanctot et al., 2017; Marris et al., 2021). One common approach, *self-play*, is to have a learning algorithm train with copies of itself as the other agents. If the algorithm is a no-$\Phi$-regret algorithm, the learned behavior will converge to a mediated equilibrium; this gives a nice characterization of the convergence behavior of the algorithm.

However, in general the strategies in a mediated equilibrium are correlated with each other. This means that in order to deploy a strategy learned in self-play, an agent must first extract it by marginalizing out other agent's strategies. This new *marginal strategy* can then be played against new agents with whom the agent did not train (and thus correlate).

**Definition 3.1** (Marginal strategy). Given some mediated equilibrium $(\mu, (\Phi_i)_{i=1}^N)$, let $s_i^\mu$ be the *marginal strategy* for $i$, where $s_i^\mu(\rho_i) \doteq \sum_{\rho_{-i} \in P_{-i}} \mu(\rho_i, \rho_{-i})$. Let $s^\mu$ be a *marginal strategy profile*, where each $\forall i \in N$ plays $s_i^\mu$.

Once a strategy has been extracted via marginalization, learning can either continue with the new agents (and potentially re-correlate), or the strategy can remain fixed.[2] We focus on the case where the strategy remains fixed. In doing so we can guarantee the performance of this strategy if learning stops, but also the show guarantees about the initial performance of a strategy that continues to learn; this is especially important in safety-critical domains.

Given a marginal strategy $s_i^\mu$, we can bound its underperformance against new agents that behave differently from the (decorrelated) training opponents by a quantity which we call *vulnerability*.

**Definition 3.2** (Vulnerability). The vulnerability of a strategy profile $s$ for player $i$ with respect to $S'_{-i} \subseteq S_{-i}$ is

$$\mathrm{Vul}_i \left( s, S'_{-i} \right) \doteq u_i(s) - \min_{s'_{-i} \in S'_{-i}} u_i(s_i, s'_{-i}).$$

Vulnerability gives a measure of how much worse $s$ will perform with new agents compared to its training performance under pessimistic assumptions—that $-i$ play the strategy profile in $S'_{-i}$ that is worst for $i$. We assume that $-i$ are not able to correlate their strategies.

Thus, if a marginal strategy profile $s^\mu$ is learned through self-play and $\mathrm{Vul}_i \left( s^\mu, S'_{-i} \right)$ is small, then $s_i^\mu$ performs roughly as well against new agents $-i$ playing some strategy profile in $S'_{-i}$. $S'_{-i}$ is used to encode assumptions about the strategies of opponents. $S'_{-i} = S_{-i}$ means opponents could play *any* strategy, but we could also set $S'_{-i}$ to be the set of strategies learnable through self-play if we believe that opponents would also be using self-play as a training procedure.

---

[2] The strategy learned in self-play prior to online learning has been called a blueprint strategy (Brown & Sandholm, 2019).

Some games have properties that make the vulnerability low. For example, in two-player constant-sum games the marginal strategies learned in self-play generalize well to new opponents since any Nash equilibrium strategy is also a maxmin strategy (von Neumann, 1928).

# 4 Guarantees via Polymatrix Decomposability

Multiplayer games are fundamentally more complex than two-player constant-sum games (Daskalakis & Papadimitriou, 2005; Daskalakis et al., 2009). However, certain multiplayer games can be decomposed into a graph of two-player games, where a player's payoffs depend only on their actions and the actions of players who are neighbors in the graph (Bergman & Fokin, 1998). In these *polymatrix* games (a subset of graphical games (Kearns et al., 2013)) Nash equilibria can be computed efficiently if player's utilities sum to a constant (Cai & Daskalakis, 2011; Cai et al., 2016).

**Definition 4.1** (Polymatrix game). A *polymatrix game* $G = (N, E, \mathrm{P}, u)$ consists of a set $N$ of players, a set of edges $E$ between players, a set of pure strategy profiles $\mathrm{P}$, and a set of utility functions $u = \{u_{ij}, u_{ji} \mid \forall (i,j) \in E\}$ where $u_{ij}, u_{ji} : \mathrm{P}_i \times \mathrm{P}_j \to \mathbb{R}$ are utility functions associated with the edge $(i, j)$ for players $i$ and $j$, respectively.

We refer to the normal-form *subgame* between $(i, j)$ as $G_{ij} = (\{i, j\}, \mathrm{P}_i \times \mathrm{P}_j, (u_{ij}, u_{ji}))$. We use $u_i$ to denote the *global utility function* $u_i : \mathrm{P} \to \mathbb{R}$ where $u_i(\rho) = \sum_{(i,j) \in E} u_{ij}(\rho_i, \rho_j)$ for each player. We use $E_i \subseteq E$ to denote the set of edges where $i$ is a player and $|E_i|$ to denote the number of such edges.

**Definition 4.2** (Constant-sum polymatrix). We say a polymatrix game $G$ is *constant-sum* if for some constant $c$ we have that $\forall \rho \in \mathrm{P}, \sum_{i \in N} u_i(\rho) = c$.

Constant-sum polymatrix (CSP) games have the desirable property that all CCE factor into a product distribution; i.e., are Nash equilibria (Cai et al., 2016). We give a relaxed version:

**Proposition 4.3.** *If $\mu$ is an $\epsilon$-CCE of a CSP game $G$, $s^{\mu}$ is an $n\epsilon$-Nash of $G$.*

This means no-external-regret learning algorithms will converge to Nash equilibria, and thus do not require a mediator to enable the equilibrium. However, they do not necessarily have the property of two-player constant-sum games that all (marginal) equilibrium strategies are maxmin strategies (Cai et al., 2016). Thus Nash equilibrium strategies in CSP games have no vulnerability guarantees. Cai et al. (2016) show that CSP games that are constant sum in each subgame are no more or less general than CSP games that are constant sum globally, since there exists a payoff preserving transformation between the two sets. For this reason we focus on CSP games that are constant sum in each subgame without loss of generality. Note that the constant need not be the same in each subgame.

## 4.1 Vulnerability on a Simple Constant-Sum Polymatrix Game

We next demonstrate why CSP games do not have bounded vulnerability on their own without additional properties. Consider the simple 3-player CSP game called Offense-Defense (Figure 2a). There are 3 players: 0, 1 and 2. Players 1 and 2 have the option to either attack 0 ($a_0$) or attack the other (e.g. $a_1$); player 0, on the other hand, may either relax ($r$) or defend ($d$). If either 1 or 2 attacks the other while the other is attacking 0, the attacker gets $\beta$ and the other gets $-\beta$ in that subgame. If both 1 and 2 attack 0, 1 and 2 get 0 in their subgame and if they attack each other, their attacks cancel out and they both get 0. If 0 plays $d$, they defend and will always get 0. If they relax, they get $-\beta$ if they are attacked and 0 otherwise. Offense-Defense is a CSP game, so any CCE is a Nash equilibrium.

Note that $\rho = (r, a_2, a_1)$ is a Nash equilibrium. Each $i \in \{1, 2\}$ are attacking the other $j \in \{1, 2\} \setminus \{i\}$, so has expected utility of 0. Deviating to attacking 0 would leave them open against the other, so $a_0$ is not a profitable deviation, as it would also give utility 0. Additionally, 0 has no incentive to deviate to $d$, since this would also give them a utility of 0.

However, $\rho$ is not a Nash equilibrium of the subgames—all $i \in \{1, 2\}$ have a profitable deviation in their subgame against 0, which leaves 0 vulnerable in that subgame. If 1 and 2 were to both deviate to $a_0$, and 0 continues to play their Nash equilibrium strategy of $r$, 0 would lose $2\beta$ utility from their equilibrium value; in other words, the vulnerability of player 0 is $2\beta$.

## 4.2 Subgame Stability

However, some constant-sum polymatrix games *do* have have bounded vulnerability; we call these *subgame stable games*. In subgame stable games, global equilibria imply equilibria at each pairwise subgame.

**Definition 4.4** (Subgame stable profile). Let $G$ be a polymatrix game with global utility functions $(u_i)_{i \in N}$. We say a strategy profile $s$ is $\gamma$-*subgame stable* if $\forall (i,j) \in E$, we have $(s_i, s_j)$ is a $\gamma$-Nash of $G_{ij}$; that is $u_{ij}(\rho_i, s_j) - u_{ij}(s_i, s_j) \leq \gamma \quad \forall \rho_i \in \mathrm{P}_i$ and $u_{ji}(\rho_j, s_i) - u_{ji}(s_j, s_i) \leq \gamma \quad \forall \rho_j \in \mathrm{P}_j$

For example, in Offense-Defense, $(r, a_2, a_1)$ is $\beta$-subgame stable; it is a Nash equilibrium but is a $\beta$-Nash of the subgame between 0 and 1 and the subgame between 0 and 2.

**Definition 4.5** (Subgame stable game). Let $G$ be a polymatrix game. We say $G$ is $(\epsilon, \gamma)$-*subgame stable* if for *any* $\epsilon$-Nash equilibrium $s$ of $G$, $s$ is $\gamma$-subgame stable.

Subgame stability connects the global behavior of play ($\epsilon$-Nash equilibrium in $G$) to local behavior in a subgame ($\gamma$-Nash in $G_{ij}$). If a polymatrix game is both constant-sum and is $(0, \gamma)$-subgame stable then we can bound the vulnerability of any marginal strategy.

**Theorem 4.6.** *Let $G$ be a CSP game. If $G$ is $(0, \gamma)$-subgame stable, then for any player $i \in N$ and CCE $\mu$ of $G$, we have* $\mathrm{Vul}_i\left(s^\mu, S_{-i}\right) \leq |E_i|\gamma$.

Theorem 4.6 tells us that using self-play to compute a marginal strategy $s^\mu$ on constant-sum polymatrix games will have low vulnerability against worst-case opponents if $\gamma$ is low. Thus, these are a set of multiplayer games where self-play is an effective training procedure.

**Proof idea.** Since $G$ is a CSP game, $s^\mu$ is a Nash equilbrium. Since $G$ is $(0, \gamma)$-subgame stable, $(s_i^\mu, s_j^\mu)$ is a $\gamma$-Nash equilibrium in each subgame, which bounds the vulnerability in each subgame. This is because

$$\min_{s_{-i} \in S_{-i}} u_i(s_i^\mu, s_{-i}) = \sum_{(i,j) \in E_i} \min_{s_j \in S_j} u_{ij}(s_i^\mu, s_j)$$

since players $j \neq i$ can minimize $i$'s utility without coordinating, as $G$ is a polymatrix game.

We give an algorithm for finding the minimum value of $\gamma$ such that a CSP game is $(0, \gamma)$-subgame stable in Appendix D.1.

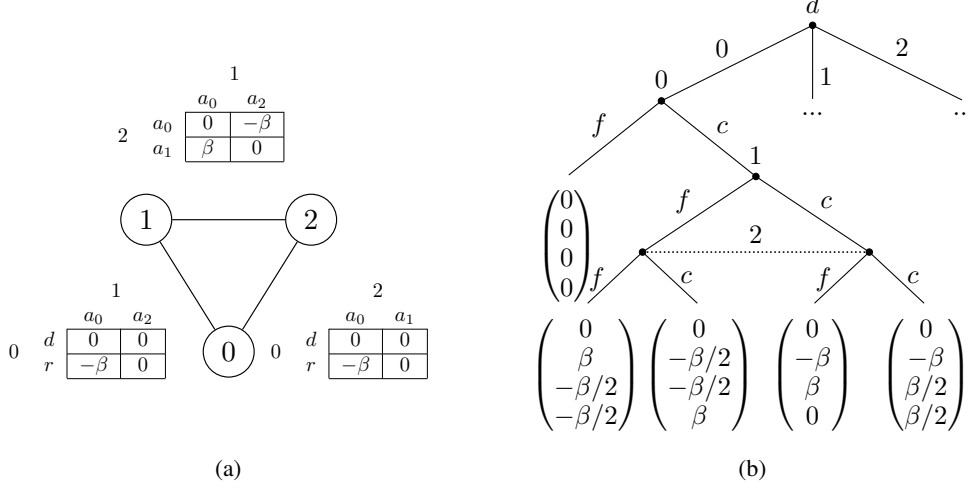

(a)  (b)

Figure 2: (a) Offense-Defense, a simple CSP game. We only show payoffs for the row player, column player payoffs are zero minus the row player's payoffs. (b) Bad Card: a game that is not overall CSP, but the subset of strategies learnable by self-play are. At the terminals, we show the dealers utility first, followed by players 0, 1 and 2, respectively.

### 4.3 Approximate Constant-Sum Polymatrix Games

Most games are not factorizable into CSP games. However, we can take any game $G$ and project it into the space of CSP games.

**Definition 4.7** ($\delta$-constant sum polymatrix). *A game $G$ is $\delta$-constant sum polymatrix ($\delta$-CSP) if there exists a CSP game $\check{G}$ with global utility function $\check{u}$ such that $\forall i \in N, \rho \in \mathrm{P}, |u_i(\rho) - \check{u}_i(\rho)| \leq \delta$. Given $G$, we denote the set of such CSP games as $\mathrm{CSP}_\delta(G)$.*

**Proposition 4.8.** *In a $\delta$-CSP game $G$ the following hold.*

1. *Any CCE of $G$ is a $2\delta$-CCE of any $\check{G} \in \mathrm{CSP}_\delta(G)$.*

2. *The marginal strategy profile of any CCE of $G$ is a $2n\delta$-Nash equilibrium of any $\check{G} \in \mathrm{CSP}_\delta(G)$.*

3. *The marginal strategy profile of any CCE of $G$ is a $2(n+1)\delta$-Nash equilibrium of $G$.*

From (3) we have that the removal of the mediator impacts players utilities by a bounded amount in $\delta$-CSP games. We give a linear program in Appendix D.2 that will find the minimum $\delta$ such that $G$ is $\delta$-CSP and returns a CSP game $\check{G} \in \mathrm{CSP}_\delta(G)$.

Combining $\delta$-CSP with $(\epsilon, \gamma)$-subgame stability lets us bound vulnerability in *any* game.

**Theorem 4.9.** *If $G$ is $\delta$-CSP and $\exists \check{G} \in \mathrm{CSP}_\delta(G)$ that is $(2n\delta, \gamma)$-subgame stable and $\mu$ is a CCE of $G$, then*

$$\mathrm{Vul}_i\left(s^\mu, S_{-i}\right) \leq |E_i|\gamma + 2\delta \leq (n-1)\gamma + 2\delta.$$

Theorem 4.9 shows that games which are close to the space of subgame stable CSP (SS-CSP) games are cases where the marginal strategies learned through self-play have bounded worst-case performance. This makes them suitable for any no-external-regret learning algorithm.

## 5 Vulnerability Against Other Self-Taught Agents

Theorem 4.9 bounds vulnerability in worst-case scenarios, where $-i$ play any strategy profile to minimize $i$'s utility. In reality, however, each player $j \in -i$ has their own interests and would only play a strategy that is reasonable under these own interests. In particular, what if each agent were also determining their own strategy via self-play in a separate training instance. How much utility can $i$ guarantee themselves in this setup?

While no-external-regret learning algorithms converge to the set of CCE, other assumptions can be made with additional information about the type of regret being minimized. For example, no-external-regret learning algorithms will play strictly dominated strategies with vanishing probability and CFR will play dominated actions with vanishing probability (Gibson, 2014). These refinements can tighten our bounds, since the part of the game that no-regret learning algorithms converge to might be closer to a CSP game than the game overall.

Consider the game shown in Figure 2b, called "Bad Card". The game starts with each player except the dealer putting $\beta/2$ into the pot. A dealer player $d$—who always receives utility 0 regardless of the strategies of the other players—then selects a player from $\{0, 1, 2\}$ to receive a "bad card", while the other two players receive a "good card". The player who receives the bad card has an option to fold, after which the game ends and all players receive their ante back. Otherwise if this player calls, the other two players can either fold or call. The pot of $\beta$ is divided among the players with good cards who call. If one player with a good card calls, they win the pot of $\beta$. If both good card players call then they split the pot. If both players with good cards fold, then the player with the bad card wins the pot.

As we shall soon show, Bad Card does not have a CSP decomposition—in fact it does not have *any* polymatrix decomposition. Since Bad Card is an extensive-form game without chance, each pure strategy profile leads to a single terminal history. Let $\mathrm{P}(z)$ be the set of pure strategy profiles that play to a terminal $z$. In order for Bad Card to be polymatrix, we would need to find subgame utility functions such that $\forall \rho \in \mathrm{P}, u_0(\rho) = u_{0,d}(\rho_0, \rho_d) + u_{0,1}(\rho_0, \rho_1) + u_{0,2}(\rho_0, \rho_2)$. Equivalently, we

could write $\forall z \in Z, \rho \in \mathrm{P}(z), u_0(z) = u_{0,d}(\rho_0, \rho_d) + u_{0,1}(\rho_0, \rho_1) + u_{0,2}(\rho_0, \rho_2)$ where $Z$ is the set of terminals. A subset of these constraints results in an infeasible system of equations.

Consider the terminals in the subtree shown in Figure 2b: $z^1 = (0, c, c, c)$, $z^2 = (0, c, c, f)$, $z^3 = (0, c, f, c)$ and $z^4 = (0, c, f, f)$. Let $\rho_i^c$ be any pure strategy that plays $c$ in this subtree and $\rho_i^f$ be any strategy that plays $f$ in this subtree for player $i$. In order for Bad Card to decompose into a polymatrix game we would need to solve the following infeasible system of linear equations:

$$u_0(z^1) = u_{0,d}(\rho_0^c, 0) + u_{0,1}(\rho_0^c, \rho_1^c) + u_{0,2}(\rho_0^c, \rho_2^c) = -\beta$$
$$u_0(z^2) = u_{0,d}(\rho_0^c, 0) + u_{0,1}(\rho_0^c, \rho_1^c) + u_{0,2}(\rho_0^c, \rho_1^f) = -\beta$$
$$u_0(z^3) = u_{0,d}(\rho_0^c, 0) + u_{0,1}(\rho_0^c, \rho_1^f) + u_{0,2}(\rho_0^c, \rho_2^c) = -\beta$$
$$u_0(z^4) = u_{0,d}(\rho_0^c, 0) + u_{0,1}(\rho_0^c, \rho_1^f) + u_{0,2}(\rho_0^c, \rho_2^f) = \beta$$

Thus, Bad Card is not a CSP game, although it is a $\beta$-CSP game. However, if we prune out dominated actions (namely, those in which a player folds after receiving a good card), the resulting game is indeed a $0$-CSP game.

Let $\mathcal{M}(\mathcal{A})$ be the set of mediated equilibria than an algorithm $\mathcal{A}$ converges to in self-play. For example, if $\mathcal{A}$ is a no-external-regret algorithm, $\mathcal{M}(\mathcal{A})$ is the set of CCE without strictly dominated strategies in their support. Let $S(\mathcal{A}) \doteq \{s^\mu \mid (\mu, (\Phi_i)_{i \in N}) \in \mathcal{M}(\mathcal{A})\}$ be the set of marginal strategy profiles of $\mathcal{M}(\mathcal{A})$, and let $S_i(\mathcal{A}) \doteq \{s_i \mid s \in S(\mathcal{A})\}$ be the set of $i$'s marginal strategies from $S(\mathcal{A})$.

Now, consider if each player $i$ learns with their own self-play algorithm $\mathcal{A}_i$. Let $\mathcal{A}_N \doteq (\mathcal{A}_1, ...\mathcal{A}_n)$ be the profile of learning algorithms, then let $S^\times(\mathcal{A}_N) \doteq \times_{i \in N} S_i(\mathcal{A}_i)$ and $S_{-i}^\times(\mathcal{A}_N) \doteq \times_{j \in -i} S_j(\mathcal{A}_j)$. Summarizing, if each player learns with a no-$\Phi_i$-regret learning algorithm $\mathcal{A}_i$, they will converge to the set of $\mathcal{M}(\mathcal{A}_i)$ equilibria. The set of marginal strategies from this set of equilibria is $S_i(\mathcal{A}_i)$ and the set of marginal strategy profiles is $S(\mathcal{A}_i)$. If each player plays a (potentially) different learning algorithm, $S^\times(\mathcal{A}_N)$ is the set of possible joint match-ups if each player plays a marginal strategy from their own algorithm's set of equilibria and $S_{-i}^\times(\mathcal{A}_N)$ is the set of profiles for $-i$.

**Definition 5.1.** We say a game $G$ is $\delta$-*CSP in the neighborhood of* $S' \subseteq S$ if there exists a CSP game $\check{G}$ such that $\forall s \in S'$ we have $|u_i(s) - \check{u}_i(s)| \leq \delta$. We denote the set of such CSP games as $\mathrm{CSP}_\delta(G, S')$.

**Definition 5.2.** We say a polymatrix game $G$ is $\gamma$-*subgame stable in the neighborhood of* $S'$ if $\forall s \in S', \forall (i, j) \in E$ we have that $(s_i, s_j)$ is a $\gamma$-Nash of $G_{ij}$.

These definitions allow us to prove the following generalization of Theorem 4.9.

**Theorem 5.3.** *For any* $i \in N$, *if* $G$ *is* $\delta$-*CSP in the neighborhood of* $S^\times(\mathcal{A}_N)$ *and* $\exists \check{G} \in \mathrm{CSP}_\delta(G, S^\times(\mathcal{A}_N))$ *that is* $\gamma$-*subgame stable in the the neighborhood of* $S(\mathcal{A}_i)$ *, then for any* $s \in S(\mathcal{A}_i)$

$$\mathrm{Vul}_i\left(s, S_{-i}^\times(\mathcal{A}_N)\right) \leq |E_i|\gamma + 2\delta \leq (n-1)\gamma + 2\delta.$$

An implication of Theorem 5.3 is that if agents use self-play to compute a marginal strategy from some mediated equilibrium and there is an SS-CSP game that is close to the original game for these strategies, then this is sufficient to bound vulnerability against strategies learned in self-play.

## 6 Computing an SS-CSP Decomposition in a Neighborhood

How might one determine if a game is well-approximated by an SS-CSP game? In addition to the algorithms presented in Appendix D, we give an algorithm, *SGDecompose*, that finds an SS-CSP decomposition for a game in a given neighborhood of strategy profiles. Pseudocode is given in Algorithm 1. SGDecompose could be used to test whether a game is well-approximated by an SS-CSP game before potentially analytically showing this property holds. We will use this algorithm in the following section to decompose Leduc poker.

As input, SGDecompose receives a neighborhood of strategies $S'$ and the set of match-ups between strategies in $S'$, given by $S^\times \doteq \times_{i \in N} S_i'$. The idea is to compute a CSP game $\check{G}$ that minimizes

a loss function with two components: how close $\check{G}$ is to $G$ in the neighborhood of $S^{\times}$ and how subgame stable $\check{G}$ is for the neighborhood of $S'$. First, $\mathcal{L}^{\delta}$ is the error between the utility functions of $G$ and $\check{G}$ ($u$ and $\check{u}$, respectively); it is a proxy for $\delta$ in $\delta$-CSP. The loss for a single strategy profile $s$ is

$$\mathcal{L}^{\delta}\left(s; \check{u}, u\right) \doteq \sum_{i \in N} |\check{u}_i(s) - u_i(s)|.$$

The other component of the overall loss function, $\mathcal{L}^{\gamma}$, measures the subgame stability. First, we define $\mathcal{L}^{\gamma}_{ij}$, which only applies to a single subgame. Let $s_{ij} = (s_i, s_j)$ be a profile for a subgame and $s^*_{ij} = (s^*_i, s^*_j)$ is a profile of deviations for that subgame. The $\mathcal{L}^{\gamma}_{ij}$ loss for this subgame is

$$\mathcal{L}^{\gamma}_{ij}(s_{ij}, s^*_{ij}; \check{u}) \doteq \max\left(\check{u}_{ij}(s^*_i, s_j) - \check{u}_{ij}(s_{ij}), 0\right) + \max\left(\check{u}_{ji}(s_i, s^*_j) - \check{u}_{ji}(s_{ij}), 0\right).$$

Then, given a strategy profile $s$ and deviation profile $s^*$ for *all* players $N$, we have

$$\mathcal{L}^{\gamma}\left(s, s^*; \check{u}\right) \doteq \sum_{(i,j) \in E} \mathcal{L}^{\gamma}_{ij}(s_{ij}, s^*_{ij}; \check{u}).$$

SGDecompose repeats over a number of epoches $T$. At the start of an epoch, we compute a best-response (for example, via sequence-form linear programming in extensive-form games) to each strategy $s'_i$ in $S'$ in each subgame; the full process is shown in Algorithm 4 in the appendix. After computing these best-responses for the current utility function of $\check{G}$, SGDecompose fits $\check{u}$ to be nearly CSP in the neighborhood of $S^{\times}$ and subgame stable in the neighborhood of $S'$. Since $S^{\times}$ is exponentially larger than $S'$, we partition it into batches, then use batch gradient descent.[3]

We use the following batch loss function, which computes the average values of $\mathcal{L}^{\delta}$ and $\mathcal{L}^{\gamma}$ over the batch then weights the losses with $\lambda$. Let $S^b$ denote a batch of strategy profiles from $S^{\times}$ with size $B$,

$$\mathcal{L}(S^b, S', S^*; \check{u}, u) \doteq \frac{\lambda}{B} \sum_{s \in S^b} \mathcal{L}^{\delta}(s; \check{u}, u) + \frac{(1-\lambda)}{|S'|} \sum_{s \in S'} \sum_{s^* \in S^*} \mathcal{L}^{\gamma}(s, s^*; \check{u}).$$

We use this loss function to update $\check{u}$, which is guaranteed to a be a valid utility function for a CSP game via its representation, see Appendix F for details. In Appendix F, we give the procedure in terms of a more efficient representation of polymatrix decompositions for extensive-form games, which we call *poly-EFGs*; which we describe in Appendix E.

---

**Algorithm 1** SGDecompose

---

**Input:** $G$, $S'$, hyperparameters $\eta$, $T$, $\lambda$, $B$
Initialize $\check{u}$ to all $0$
$S^{\times} \leftarrow \bigtimes_{i \in N} \hat{S}_i$
**for** $t \in 1...T$ **do**
    $S^* \leftarrow \text{getBRs}(\check{G}, S')$
    $\mathcal{B} \leftarrow$ partition of $S^{\times}$ into batches of size $B$
    **for** $S^b \in \mathcal{B}$ **do**
        $g \leftarrow \nabla_{\check{u}} \mathcal{L}(S^b, S', S^*; \check{u}, u)$
        $\check{u} \leftarrow \check{u} - \eta \cdot \frac{g}{\|g\|_2}$ {update $\check{u}$ using normalized gradient; this helps with stability}
    **end for**
**end for**
{Lastly, output $\delta$ and $\gamma$}
$\delta \leftarrow \max_{s \in S^{\times}} |u_i(s) - \check{u}_i(s)|$
$\gamma \leftarrow \max_{s \in S'} \max_{i \neq j \in N \times N} \left(\check{u}_{ij}(BR_{ij}(s_j), s_j) - \check{u}_{ij}(s_i, s_j)\right)$
**return** $\check{u}, \gamma, \delta$

---

## 7 Experiments

Approaches using regret-minimization in self-play have been shown to outperform expert human players in some multiplayer games, the most notable example being multiplayer no-limit Texas hold 'em (Brown & Sandholm, 2019), despite no formal guarantees.

---

[3]One could also partition $S'$ into batches if it were too large.

**Conjecture 7.1.** Self-play with regret minimization performs well in multiplayer Texas hold 'em because "good" players (whether professional players or strategies learned by self-play) play in a part of the games' strategy space that is close to an SS-CSP game (i.e. low values of $\gamma, \delta$).

While multiplayer no-limit Texas hold 'em is too large to directly check the properties developed in this work, we use a smaller poker game, called Leduc poker (Southey et al., 2012), to suggest why regret-minimization "works" in multiplayer Texas hold 'em. Leduc poker was originally developed for two players but was extended to a 3-player variant by Abou Risk & Szafron (2010); we use the 3-player variant here. The game has 8 cards, two rounds of betting, one private and one public card.

We first use a self-play algorithm to learn strategies, then use SGDecompose to see if this part of Leduc Poker is close to an SS-CSP game. We give a summary of results here, please refer to Appendix G for full details. We use CFR+ (Tammelin, 2014; Tammelin et al., 2015) as a self-play algorithm to compute a set of approximate marginal strategies.[4] CFR+ was chosen because of its improved efficiency over CFR. CFR+ is a deterministic algorithm, so we use different random initializations of CFR+'s initial strategy in order to generate a set of CCE. We will use 30 runs of CFR+ as input to SGDecompose; and have 30 runs of SGDecompose (i.e. we trained CFR+ 900 times in total in self-play). This will give us a value of $\delta$ and $\gamma$ for each run.

We found that Leduc poker was well-approximated by an SS-CSP game in the neighborhood of strategies learned by CFR+. In particular, across runs, Leduc poker was on average (with standard errors) $\delta = 0.009 \pm 0.00046$-CSP and $\gamma = 0.004 \pm 0.00016$-subgame stable in the neighborhood of CFR+-learned strategies. How well do these values bound vulnerability with respect to other CFR-learned strategies? For each of the runs, we computed the vulnerability with respect to the strategies of that run, by evaluating each strategy against each other and taking the maximum vulnerability. We compare these values to the upper bounds implies by the values of $\delta$ and $\gamma$ for each run and Theorem 5.3. We found the computed values of $\delta$ and $\gamma$ do a good job of upper bounding the vulnerability. Across the runs, the bounds are at minimum $1.89$ times the vulnerability, at maximum $3.05$ times the vulnerability and on average $2.51$ times as large, with a standard error of $0.049$.

We repeated these experiments with a toy hanabi game—where strategies learned in self-play are highly vulnerable—which we found to have much higher values of $\delta$ and $\gamma$; details are in Appendix H.

It was previously believed that CFR does not compute an $\epsilon$-Nash equilibrium on 3-player Leduc for any reasonable value of $\epsilon$. Abou Risk & Szafron (2010) found CFR computed a $0.130$-Nash equilibrium. We found that CFR+ always computed an approximate Nash equilbrium with $\epsilon \leq 0.013$. Appendix G.2 shows that CFR also computes an approximate Nash equilbrium with $\epsilon \leq 0.017$.

## 8 Conclusion

Self-play has been incredibly successful in producing strategies that perform well against new opponents in two-player constant-sum games. Despite a lack of theoretical guarantees, self-play seems to also produce good strategies in some multiplayer games (Brown & Sandholm, 2019). We identify a structural property of multiplayer, general-sum game that allow us to establish guarantees on the performance of strategies learned via self-play against new opponents. We show that any game can be projected into the space of constant-sum polymatrix games, and if there exists a game with this set with high subgame stability (low $\gamma$), strategies learned through self-play have bounded loss of performance against new opponents.

We conjecture that Texas hold 'em is one such game. We investigate this claim on Leduc poker, and find that CFR+ plays strategies from a part of the strategy space in Leduc poker that is well-approximated by a subgame stable constant-sum polymatrix game. This work lays the groundwork for guarantees for self-play in multiplayer games. However, there is room for algorithmic improvement and efficiency gains for checking these properties in very large extensive-form games.

---

[4]We use the OpenSpiel implementation (Lanctot et al., 2019).

## Acknowledgements

Computation for this work was provided by the Digital Research Alliance of Canada. Revan MacQueen was supported by Alberta Innovates and NSERC during completion of this work. This work was funded in part by an NSERC Discovery Grant. James R. Wright holds a Canada CIFAR AI Chair through the Alberta Machine Intelligence Institute. Thank you to Dustin Morrill, Michael Bowling and Nathan Sturtevant for helpful conversations and feedback, and the anonymous reviewers for valuable comments.

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

# A  Marginals of a CCE May Not Be a CCE

|       | $a$        | $b$        |
|-------|------------|------------|
| $a$   | $1, 0$     | $-1, -1$   |
| $a$   | $-1, -1$   | $0, 1$     |

Figure 3: The marginal strategies of a CCE do not generally form a CCE themselves.

Here we give an example showing the marginal strategies of a CCE may not form a CCE. Consider $\mu$ s.t. $\mu(a, a) = 0.5$ and $\mu(b, b) = 0.5$. $\mu$ is a CCE. $\mathbb{E}_{\rho \sim \mu}[u_i(\rho)] = 0.5$ for each player. If row ($r$) were to player $a$ and column continues to play according to $\mu$, row's utility is 0; if $r$ plays $b$ instead, their utility is now $-0.5$. Thus $r$ has no profitable deviations from the CCE recommendations. Column does not either, this can be shown with a symmetric argument.

Row's marginal strategy $s_r^\mu$ plays $a$ with probability 0.5 and $b$ with probability 0.5, $s_c^\mu$ does likewise. $u_r(s_r^\mu, s_c^\mu) = u_c(s_r^\mu, s_c^\mu) = -0.25$. However, $a$ is a profitable deviation for $r$ now since $0 > -0.25$, thus the decorrelated strategies from the same CCE are also not a CCE.

# B  Hindsight Rationality With Respect to Action Deviations Does Not Imply Nash

Here we show that hindsight rationality with respect to action deviations does not imply Nash equilibrium in 2 player constant-sum games. We show this with a 1 player game. Consider the agents strategy, shown in blue, which receives utility of 1. Deviating to $[I_1 : b, I_2 : a]$ will increase the player's utility to 2, so the blue strategy is not a Nash equilibrium. However, this would require two simultaneous action deviations, one at $I_1$ to $b$ and one at $I_2$ to $a$. Neither of these deviations increases the player's utility on their own, so the player is hindsight rational w.r.t. action deviations.

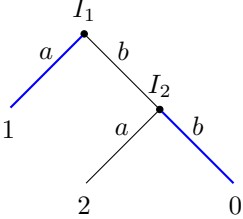

Figure 4: Action deviations in a simple game.

# C  Omitted Proofs

**Proposition C.1** (Cai et al. (2016))**.** *In CSP games, for any CCE $\mu$, if $i$ deviates to $s_i$, then their expected utility if other players continue to play $\mu$ is equal to their utility if other player where to play the marginal strategy profile $s_{-i}^\mu$:*

$$\mathbb{E}_{\rho \sim \mu}[u_i(s_i, \rho_{-i})] = u_i(s_i, s_{-i}^\mu) \; \forall s_i \in S_i$$

## C.1  Proof of Proposition 4.3

**Proposition 4.3.** *If $\mu$ is an $\epsilon$-CCE of a CSP game $G$, $s^\mu$ is an $n\epsilon$-Nash of $G$.*

*Proof.* Since $\mu$ is an $\epsilon$-CCE, $\forall i \in N$, we have

$$\max_{\rho_i' \in \mathrm{P}_i} \mathbb{E}_{\rho \sim \mu}\left[u_i(\rho_i', \rho_{-i})\right] - \mathbb{E}_{\rho \sim \mu}\left[u_i(\rho)\right] \leq \epsilon$$

which implies (by Proposition C.1) that $\forall i \in N$,

$$\max_{\rho_i' \in P_i} u_i(\rho_i', s_{-i}^\mu) - \mathbb{E}_{\rho \sim \mu} [u_i(\rho)] \leq \epsilon$$

$$\implies \max_{\rho_i' \in P_i} u_i(\rho_i', s_{-i}^\mu) \leq \epsilon + \mathbb{E}_{\rho \sim \mu} [u_i(\rho)].$$

Summing over $N$, we get

$$\sum_{i \in N} \max_{\rho_i' \in P_i} u_i(\rho_i', s_{-i}^\mu) \leq \sum_{i \in N} (\epsilon + \mathbb{E}_{\rho \sim \mu} [u_i(\rho)]) \tag{1}$$

$$= \sum_{i \in N} \epsilon + \sum_{i \in N} \mathbb{E}_{\rho \sim \mu} [u_i(\rho)] \tag{2}$$

$$= \sum_{i \in N} \epsilon + \mathbb{E}_{\rho \sim \mu} \left[ \sum_{i \in N} u_i(\rho) \right] \tag{3}$$

$$= n\epsilon + c \tag{4}$$

$$= n\epsilon + \sum_{i \in N} u_i(s^\mu). \tag{5}$$

Where (4) and (5) use the fact that $\forall \rho \in P$, $\sum_{i \in N} u_i(\rho) = c$ for some constant. The above inequalities give us

$$\sum_{i \in N} \max_{\rho_i' \in P_i} u_i(\rho_i', s_{-i}^\mu) \leq n\epsilon + \sum_{i \in N} u_i(s^\mu).$$

Rearranging, we get

$$\sum_{i \in N} \underbrace{\max_{\rho_i' \in P_i} u_i(\rho_i', s_{-i}^\mu) - u_i(s^\mu)}_{\geq 0} \leq n\epsilon.$$

All terms in the sum are are non-negative because $\rho_i'$ is a best-response to $s_{-i}^\mu$. Then any particular term in the summation is upper bounded by $n\epsilon$. $\qquad\square$

## C.2 Proof of Theorem 4.6

**Proposition C.3.** *In two-player constant-sum games, for any $\epsilon$-Nash equilibrium $s$ and player $i$, we have*

$$u_i(s) - \min_{s_{-i}' \in S_{-i}} u_i(s_i, s_{-i}') \leq \epsilon.$$

The proof of Proposition C.3 is immediate from the fact that no player can gain more than $\epsilon$ utility by deviating and the game is constant-sum.

**Theorem 4.6.** *Let $G$ be a CSP game. If $G$ is $(0, \gamma)$-subgame stable, then for any player $i \in N$ and CCE $\mu$ of $G$, we have $\mathrm{Vul}_i(s^\mu, S_{-i}) \leq |E_i|\gamma$.*

*Proof.* First we show 1. Any marginal strategy $s^\mu$ of a CCE $\mu$ is a Nash equilibrium of $G$ Cai et al. (2016). Then,

$$\mathrm{Vul}_i(s^\mu, S_{-i}) \doteq u_i(s^\mu) - \min_{s_{-i} \in S_{-i}} u_i(s_i^\mu, s_{-i})$$

$$= \sum_{(i,j) \in E_i} u_{ij}(s_i^\mu, s_j^\mu) - \min_{s_{-i} \in S_{-i}} \left( \sum_{(i,j) \in E_i} u_{ij}(s_i^\mu, s_j) \right)$$

$$= \sum_{(i,j) \in E_i} u_{ij}(s_i^\mu, s_j^\mu) - \sum_{(i,j) \in E_i} \min_{s_j \in S_j} u_{ij}(s_i^\mu, s_j).$$

Where the last line uses the fact that $-i$ minimize $i's$ utility, so can do so without coordinating since $G$ is polymatrix. Continuing,

$$= \sum_{(i,j) \in E_i} \left( u_{ij}(s_i^\mu, s_j^\mu) - \min_{s_j \in S_j} u_i(s_i^\mu, s_j) \right)$$

$$\leq \sum_{(i,j) \in E_i} \gamma$$

$$\leq |E_i| \gamma,$$

where by $(0, \gamma)$-subgame stability of each $G_{ij}$, $(s_i^\mu, s_i^\mu)$ is a $\gamma$-Nash of $G_{ij}$. By Proposition C.3, we have $u_{ij}(s_i^\mu, s_j^\mu) - \min_{s_j \in S_j} u_i(s_i^\mu, s_j) \leq \gamma$. $\qquad \square$

### C.3  Proof of Proposition 4.8

**Proposition 4.8.** *In a $\delta$-CSP game $G$ the following hold*

1. *Any CCE of $G$ is a $2\delta$-CCE of any $\check{G} \in \mathrm{CSP}_\delta(G)$.*

2. *The marginalized strategy profile of any CCE of $G$ is a $2n\delta$-Nash equilibrium of any $\check{G} \in \mathrm{CSP}_\delta(G)$.*

3. *The marginalized strategy profile of any CCE is a $2(n+1)\delta$-Nash equilibrium of $G$*

*Proof.* First we prove claim 1. Let $\check{u}_i$ denote the utility function of $i$ in $\check{G}$. Note that $\forall \rho \in \mathrm{P}$ we have $|\check{u}_i(\rho) - u_i(\rho)| \leq \delta \ \forall i \in N$. Let $\mu$ be any CCE of $G$. The definition of CCE states

$$\mathbb{E}_{\rho \sim \mu} \left[ u_i(\rho_i', \rho_{-i}) - u_i(\rho) \right] \leq 0 \quad \forall i \in N, \rho_i' \in \mathrm{P}_i.$$

It is sufficient to consider only player $i$. We can preserve the inequality by substituting $\check{u}_i(\rho_i', \rho_{-i}) - \delta$ in place of $u_i(\rho_i', \rho_{-i})$ and $\check{u}_i(\rho) + \delta$ in place of $u_i(\rho)$. This gives us

$$\mathbb{E}_{\rho \sim \mu} \left[ \check{u}_i(\rho_i', \rho_{-i}) - \delta - (\check{u}_i(\rho) + \delta) \right] \leq 0 \quad \forall \rho_i' \in \mathrm{P}_i$$

$$\implies \mathbb{E}_{\rho \sim \mu} \left[ \check{u}_i(\rho_i', \rho_{-i}) - \check{u}_i(\rho) \right] \leq 2\delta \quad \forall \rho_i' \in \mathrm{P}_i.$$

Thus claim 1 is shown. Claim 2 is an immediate corollary of claim 1 and Proposition 4.3. Lastly, we show claim 3. By claim 2, we have the marginalized strategy of $\mu$, $s^\mu$, is a $2n\delta$-Nash equilibrium of $\check{G} \in \mathrm{CSP}_\delta(G)$. That is for any $i \in N$,

$$\check{u}_i(\rho_i', s_{-i}^\mu) - \check{u}_i(s_i^\mu, s_{-i}^\mu) \leq 2n\delta \quad \forall \rho_i' \in \mathrm{P}_i.$$

However, since $G$ is $\delta$-CSP, we may substitute $u_i(\rho_i', s_{-i}^\mu) - \delta$ in place of $\check{u}_i(\rho_i', s_{-i}^\mu)$ and $u_i(s_i^\mu, s_{-i}^\mu) + \delta$ in place of $\check{u}_i(s_i^\mu, s_{-i}^\mu)$ as preserve the inequality.

$$\left( u_i(\rho_i', s_{-i}^\mu) - \delta \right) - \left( u_i(s_i^\mu, s_{-i}^\mu) + \delta \right) \leq 2n\delta \quad \forall \rho_i' \in \mathrm{P}_i.$$

Rearranging, this gives us

$$u_i(\rho_i', s_{-i}^\mu) - u_i(s_i^\mu, s_{-i}^\mu) \leq 2n\delta + 2\delta = 2(n+1)\delta \quad \forall \rho_i' \in \mathrm{P}_i.$$

$\qquad \square$

### C.4  Proof of Theorem 4.9

**Theorem 4.9.** *If $G$ is $\delta$-CSP and $\exists \check{G} \in \mathrm{CSP}_\delta(G)$ that is $(2n\delta, \gamma)$-subgame stable and $\mu$ is a CCE of $G$, then*

$$\mathrm{Vul}_i \left( s^\mu, S_{-i} \right) \leq |E_i| \gamma + 2\delta \leq (n-1)\gamma + 2\delta,$$

The proof is largely the same as Theorem 4.6, with added approximation since $G$ is no longer CSP.

*Proof.* Let $\check{G}$ be a polymatrix game that is $(2n\delta, \gamma)$-subgame stable such that $\check{G} \in \text{CSP}_\delta(G)$. Let $\check{u}_i$ denote the utility function of $i$ in $\check{G}$. By Proposition 4.8, $\mu$ is a $2n\delta$-Nash equilibrium of $\check{G}$. Then,

$$\text{Vul}_i\left(s^\mu, S_{-i}\right) \doteq u_i(s^\mu) - \min_{s'_{-i} \in S_{-i}} u_i(s^\mu_i, s'_{-i})$$

$$\leq \check{u}_i(s^\mu) - \min_{s'_{-i} \in S_{-i}} \check{u}_i(s^\mu_i, s'_{-i}) + 2\delta,$$

since $G$ is $\delta$-CSP. Then expanding $\check{u}_i$ across $i$'s subgames we have

$$\sum_{(i,j) \in E_i} \check{u}_{ij}(s^\mu_i, s^\mu_j) - \min_{s'_{-i} \in S_{-i}} \sum_{(i,j) \in E_i} \check{u}_i(s^\mu_i, s_j) + 2\delta$$

$$= \sum_{(i,j) \in E_i} \check{u}_{ij}(s^\mu_i, s^\mu_j) - \sum_{(i,j) \in E_i} \min_{s'_j \in S_j} \check{u}_i(s^\mu_i, s'_j) + 2\delta.$$

Where, as in Theorem 4.6, the last line uses the fact that $\check{G}$ is polymatrix, $G_{ij}$ is constant-sum and $-i$ minimize $i's$ utility and can do so by without coordinating. Continuing, we have

$$\sum_{(i,j) \in E_i} \check{u}_{ij}(s^\mu_i, s^\mu_j) - \sum_{(i,j) \in E_i} \min_{s'_j \in S_j} \check{u}_i(s^\mu_i, s'_j) + 2\delta$$

$$= \sum_{(i,j) \in E_i} \left( \check{u}_{ij}(s^\mu_i, s^\mu_j) - \min_{s'_j \in S_j} \check{u}_i(s^\mu_i, s'_j) \right) + 2\delta$$

$$\leq \sum_{(i,j) \in E_i} \gamma + 2\delta$$

$$= |E_i|\gamma + 2\delta$$

$$\leq (n-1)\gamma + 2\delta.$$

Where by $(2n\delta, \gamma)$-subgame stability of each $G_{ij}$, $(s^\mu_i, s^\mu_j)$ is a $\gamma$-Nash of $G_{ij}$. By Proposition C.3, $s^\mu_i$ can lose at most $\gamma$ to a worst case opponent $s'_j$ in each subgame, since $\check{G}_{ij}$ is two-player constant-sum. $\qquad\square$

## C.5 Proof of Theorem 5.3

**Theorem 5.3.** *If $G$ is $\delta$-CSP in the neighborhood of $S^\times(\mathcal{A}_N)$ and $\exists \check{G} \in \text{CSP}_\delta(G, S^\times(\mathcal{A}_N))$ that is $\gamma$-subgame stable in the the neighborhood of $S(\mathcal{A}_i)$, then for any $s \in S(\mathcal{A}_i)$*

$$\text{Vul}_i\left(s, S^\times_{-i}(\mathcal{A}_N)\right) \leq |E_i|\gamma + 2\delta \leq (n-1)\gamma + 2\delta.$$

*Proof.* The proof is very similar to Theorem 4.9. Writing the definition of vulnerability we have

$$\text{Vul}_i\left(s, S(\mathcal{A})\right) \doteq u_i(s) - \min_{s'_{-i} \in S^\times_{-i}(\mathcal{A}_N)} u_i(s, s'_{-i}), \tag{6}$$

since $G$ is $\delta$-CSP in the neighborhood of $S^\times(\mathcal{A}_N)$. Swapping out the utility of $u_i$ for $\check{u}$, we have

$$(6) \leq \check{u}_i(s) - \min_{s'_{-i} \in S^\times_{-i}(\mathcal{A}_N)} \check{u}_i(s_i, s'_{-i}) + 2\delta$$

Since $\check{G}$ is a polymatrix game,

$$\check{u}_i(s) - \min_{s'_{-i} \in S^\times_{-i}(\mathcal{A}_N)} \check{u}_i(s_i, s'_{-i}) + 2\delta \tag{7}$$

$$= \sum_{(i,j) \in E_i} \check{u}_{ij}(s_i, s_j) - \min_{s'_{-i} \in S^\times_{-i}(\mathcal{A}_N)} \sum_{(i,j) \in E_i} \check{u}_{ij}(s_i, s_j) + 2\delta \tag{8}$$

$$= \left( \sum_{(i,j) \in E_i} \check{u}_{ij}(s_i, s_j) - \min_{s'_j \in S_j(\mathcal{A}_j)} \check{u}_{ij}(s_i, s'_j) \right) + 2\delta. \tag{9}$$

Where, as in Theorem 4.6 and Theorem 4.9, the last line uses the fact that $\check{G}$ is polymatrix, $G_{ij}$ is constant-sum and $-i$ minimize $i's$ utility and can do so by without coordinating.

Since $\check{G}$ is $\gamma$-subgame stable in the neighborhood of $S(\mathcal{A}_i)$ and $s \in S(\mathcal{A}_i)$, then means $(s_i, s_j)$ is a $\gamma$-Nash for each subgame $\check{G}_{ij}$, so has bounded vulnerability within that subgame.

$$(9) \leq \left( \sum_{(i,j) \in E_i} \gamma \right) + 2\delta$$
$$\leq |E_i|\gamma + 2\delta$$
$$\leq (n-1)\gamma + 2\delta$$

$\square$

# D    Normal-Form Algorithms

## D.1    Computing Subgame Stability

Let $\underset{\sim}{\gamma}$ be the minimum $\gamma$ such that a CSP game $G$ is $(0, \gamma)$-subgame stable. How do we compute $\underset{\sim}{\gamma}$? Does it involve computing all equilibria of $G$ and checking their subgame stability? The answer is no, it can be done in polynomial time in the number of pure strategies. We next provide an algorithm for computing $\underset{\sim}{\gamma}$. The algorithm involves solving a linear program for each edge in the graph and each pure strategy of those players. This linear program takes a pure strategy $\rho_i'$, and finds a Nash equilibrium of $G$ that maximizes $i$'s incentive to deviate to $\rho_i'$ when *only* considering their utility in $G_{ij}$; call this quantity $\gamma_{ij}^{\rho_i'}$. If there are no such Nash equilibria the solver returns "infeasible". If the solver does not return "infeasible", we update $\underset{\sim}{\gamma} = \max_{(i,j) \in E_i} \max_{\rho_i' \in P_i} \gamma_{ij}^{\rho_i'}$.

---

**Algorithm 2** Compute $\gamma$

---

   **Input:** $G = (N, E, P, u)$, a polymatrix game
   $\gamma \leftarrow -\infty$
   **for** $(i, j) \in E$ **do**
      **for** $\rho_i' \in P_i$ **do**
         **if** LP1$(i, j, \rho_i')$ not infeasible **then**
            $\gamma_{ij}^{\rho_i'} \leftarrow$ LP1$(i, j, \rho_i')$
            $\gamma \leftarrow \max(\gamma, \gamma_{ij}^{\rho_i'})$
         **end if**
      **end for**
      **for** $\rho_j' \in P_j$ **do**
         **if** LP1$(j, i, \rho_j')$ not infeasible **then**
            $\gamma_{ji}^{\rho_j'} \leftarrow$ LP1$(j, i, \rho_j')$
            $\gamma \leftarrow \max(\gamma, \gamma_{ji}^{\rho_j'})$
         **end if**
      **end for**
   **end for**
   **return** $\gamma$

---

Let $a_i(\rho_i', \mu)$ be the advantage of deviating to $\rho_i'$ from a joint distribution over pure strategies:

$$a_i(\rho_i', \mu) \doteq \sum_{(i,j) \in E_i} \underbrace{u_{ij}(\rho_i', s_j^\mu)}_{(a)} - \underbrace{\mathbb{E}_{\rho \sim \mu} \left[ \sum_{(i,j) \in E_i} u_{ij}(\rho_i, \rho_j) \right]}_{(b)}$$

Note that $(a)$ is a linear function of $\mu$, since $s_j^\mu$ is a marginal strategy. $(b)$ is also a linear function of $\mu$, and so $a_i(\rho_i', \mu)$ is a linear function of $\mu$. Likewise, let

$$a_{ij}(\rho_i', \mu) \doteq u_{ij}(\rho_i', s_j^\mu) - \mathbb{E}_{\rho \sim \mu}\left[u_{ij}(\rho_i, \rho_j)\right].$$

be the advantage of $\rho_i'$ in the subgame between $i$ and $j$. $\mathrm{LP1}(i, j, \rho_i')$ is given below. The decision variables are the weights of $\mu$ for each $\rho \in \mathrm{P}$ and $\gamma_{ij}^{\rho_i'}$.

**LP 1**

$$
\begin{aligned}
\max \quad & \gamma_{ij}^{\rho_i'} \\
\text{s.t.} \quad & a_i(\rho_i, \mu) \leq 0 \quad \forall i \in N, \rho_i \in \mathrm{P}_i \\
& a_{ij}(\rho_i', \mu) \geq \gamma_{ij}^{\rho_i'} \\
& \sum_{\rho \in \mathrm{P}} \mu(\rho) = 1 \\
& \mu(\rho) \in [0, 1] \quad \forall \rho \in \mathrm{P}
\end{aligned}
$$

We can get away with computing a CCE rather than targeting Nash equilibria because the marginals of any CCE are Nash equilibria in CSP games Cai et al. (2016).

The whole procedure runs in polynomial time in the size of the game. We need to solve an LP, which takes polynomial time, at most $n^2 \max_{i \in N} |\mathrm{P}_i|$ times.

### D.2 Finding Constant-Sum Polymatrix Decomposition

Projecting a game into the space of CSP games with minimum $\delta$ can be done with a linear program. Let $\underline{\delta}$ be the minimum $\delta$ such that $G$ is $\delta$-CSP. We give a linear program that finds $\underline{\delta}$ and returns a CSP game $\check{G} \in \mathrm{CSP}_{\underline{\delta}}(G)$. The decision variables are the values of $\check{u}_{ij}(\rho)$ for all $i \neq j \in N, \rho \in \mathrm{P}$, $\underline{\delta}$ and constants for each subgame $c_{ij}$, for all $i \neq j$.

**LP 2**

$$
\begin{aligned}
\min \quad & \underline{\delta} \\
\text{s.t.} \quad & u_i(\rho) - \sum_{j \in -i} \check{u}_{ij}(\rho_i, \rho_j) \leq \underline{\delta} \quad \forall i \in N, \rho \in \mathrm{P} \\
& u_i(\rho) - \sum_{j \in -i} \check{u}_{ij}(\rho_i, \rho_j) \geq -\underline{\delta} \quad \forall i \in N, \rho \in \mathrm{P} \\
& \check{u}_{ij}(\rho_i, \rho_j) + \check{u}_{ji}(\rho_i, \rho_j) = c_{ij} \quad \forall i \neq j \in N, (\rho_i, \rho_j) \in \mathrm{P}_{ij},
\end{aligned}
$$

## E  Extensive-Form Games

Section 4 developed notions of approximately CSP and subgame stable in the context of normal-form games. Here, we apply these concepts to extensive-form games. While any extensive-form game has an equivalent induced normal-form game, analysing properties of an EFG through its induced normal is intractable for moderately-sized EFGs, since the size the normal-form representation is exponentially larger.

After background on extensive-form games, we introduce a novel "extensive-form version" of normal-form polymatrix games, which we call *poly-EFGs*. The major benefit of poly-EFGs over normal-form polymatrix games is their efficiently: poly-EFGs are exponentially more compact than an equivalent normal-form polymatrix game. The results of this section extend the theory of Sections 4 and 5 using this more efficient representation. We also give a proof-of-concept showing that poly-EFGs can be used to efficiently decompose extensive-form games by giving an algorithm for decomposing a perfect information EFG into a poly-EFG.

## E.1 Background on Extensive-Form Games

We use the imperfect information extensive-form game (EFG) as a model for sequential multi-agent strategic situations. An imperfect information extensive-form game is a 10-tuple $(N, \mathcal{A}, H, Z, A, P, u, \mathcal{I}, c, \pi_c)$ where $N$ is a set of players; $\mathcal{A}$ is a set of actions; $H$ is a set of sequences of actions, called *histories*; $Z \subseteq H$ is a set of terminal histories; $A : H \to \mathcal{A}$ is a function that maps a history to available actions; $P : H \to N$ is the player function, which assigns a player to choose an action at each non-terminal history; $u = \{u_i\}_{i \in N}$ is a set of utility functions where $u_i : Z \to \mathbb{R}$ is the utility function for player $i$; $\mathcal{I} = \{\mathcal{I}_i\}_{i \in N}$ where $\mathcal{I}_i$ is a partition of the set $\{h \in H : P(h) = i\}$ such that if $h, h' \in I \in \mathcal{I}_i$ then $A(h) = A(h')$. We call an element $I \in \mathcal{I}_i$ an information set. The chance player $c$ has a function $\pi_c(a, h) \; \forall h : P(h) = c$ which returns the probability of random nature events $a \in \mathcal{A}$. Let $N_c = N \cup \{c\}$ be the set of players including chance.

For some history $h$, the $j$th action in $h$ is written $h_j$. A sub-history of $h$ from the $j$th to $k$th actions is denoted $h_{j:k}$ and we use $h_{:k}$ as a short-hand for $h_{0:k}$. If a history $h'$ is a prefix of history $h$, we write $h' \sqsubseteq h$ and if $h'$ is a proper prefix of $h$, we write $h' \sqsubset h$.

A *pure strategy* in an EFG is a deterministic choice of actions for the player at every decision point. We use $\rho_i : \mathcal{I}_i \to \mathcal{A}$ to denote a pure strategy of player $i$, and the set of all pure strategies as $P_i$. Likewise, $s_i \in \Delta(P_i) = S_i$ is a *mixed strategy*, where $\Delta(X)$ denotes the set of probability distributions over a domain $X$.

There are an exponential number of pure strategies in the number of information sets. A *behavior strategy* is a compact representation of the behavior of an agent that assigns a probability distribution over actions to each information set. We use $\pi_i \in \Pi_i = (\Delta(A(I)))_{I \in \mathcal{I}_i}$ to denote a behavior strategy of player $i$ and $\pi_i(a, I)$ as the probability of playing action $a$ at $I$. Let $I(h)$ be the unique information set such that $h \in I$. We overload $\pi_i(a, h) = \pi_i(a, I(h))$. We use $\rho \in P$, $s \in S$ and $\pi \in \Pi$ to denote pure, mixed and behavior strategy profiles, respectively. Note that $P$ is a subset of both $\Pi$ and $S$.

Given a behavior strategy profile, let

$$p_i(h_1, h_2, \pi_i) \doteq \prod_{h_1 \sqsubseteq ha \sqsubseteq h_2, P(ha)=i} \pi_i(a, h)$$

$$p(h_1, h_2, \pi) \doteq \prod_{i \in N_c} p_i(h_1, h_2, \pi_i)$$

$$p_{-i}(h_1, h_2, \pi_{-i}) \doteq \prod_{j \in N_c \setminus \{i\}} p_j(h_1, h_2, \pi_j)$$

be the probability of transitioning from history $h_1$ to $h_2$ according to $\pi_i$, $\pi$ and $\pi_{-i}$, respectively. Let $p_i(z, \pi_i), p_{-i}(z, \pi_i)$ and $p(z, \pi_i)$ be short-hands for $p_i(\varnothing, z, \pi_i), p_{-i}(\varnothing, z, \pi_{-i})$ and $p(\varnothing, z, \pi)$, respectively, where $\varnothing$ is the empty history.

We define the utility of a behavior strategy as:

$$u_i(\pi) \doteq \mathbb{E}_{z \sim \pi} [u_i(z)] = \sum_{z \in Z} p(z, \pi) u_i(z) = \sum_{z \in Z} \left( \prod_{i \in N_c} p_i(z, \pi_i) \right) u_i(z).$$

*Perfect recall* is a common assumption made on the structure of information sets in EFGs that prevents players from forgetting information they once possessed. Formally, for any $h \in I$ let $X_i(h)$ denote the set of $(I, a)$ s.t. $I \in \mathcal{I}_i$ and $\exists h' \in I$ and $h'a \sqsubseteq h$. Let $X_{-i}(h)$ be defined analogously for $-i$ and $X(h)$ for all players.

**Definition E.1** (Perfect recall)**.** If $\forall I \in \mathcal{I}_i, \forall h, h' \in I, X_i(h) = X_i(h')$ then $i$ has perfect recall. If all players possess perfect recall in some EFG $G$, we call $G$ a game of perfect recall.

In games of perfect recall, the set of behavior strategies and mixed strategies are equivalent: any behavior strategy can be converted into a mixed strategy which is outcome equivalent over the set of terminal histories (i.e. has the same distribution over $Z$) and vice-versa.

**Theorem E.2** (Kuhn (1953))**.** *In games of perfect recall, any behavior strategy $\pi_i$ has an equivalent mixed strategy $s_i$ (and vice versa), such that*

$$p_i(z, \pi_i) = \mathbb{E}_{\rho_i \sim s_i} [p_i(z, \rho_i)].$$

Theorem E.2 establishes a connection between equilibria in behavior and mixed strategies: a Nash equilibrium behavior strategy profile implies the equivalent mixed strategy profile is also a Nash equilibrium in mixed strategies and vice-versa.

We may also reduce any extensive-form game into an equivalent normal-form game.

**Definition E.3** (Induced normal-form). The *induced normal-form* of an extensive-form game $G$ (with utility functions $u_i$) is a normal-form game $G' = (N, \mathrm{P}, u')$ such that $u_i'(\rho) = u_i(\rho)$

The induced normal-form of an EFG has players making all decisions up-front. It is not always practical to construct an induced normal-form, but the concept is useful for proving things about EFGs.

## E.2 Poly-EFGs

What is the appropriate extension of polymatrix games to EFGs? Given some $n$-player EFG $G$, what should the subgame between $i$ and $j$ be in the graph? Unlike in normal form games, players act sequentially, and $i$ and $j$ may either observe actions or have their utility impacted by other players. There is additional structure present in EFGs beyond the players' pure strategies.

The approach we take is to have an EFG for each pair of players $i$ and $j$. For simplicity, we assume that each subgame $G_{ij}'$ shares the same structure as some $n$-player game $G$, but information sets where $P(I) \notin \{i, j\}$ now belong to the chance player $c$.

**Definition E.4** (Subgame). Let $G = (N, \mathcal{A}, H, Z, A, P, u, \mathcal{I}, c, \pi_c)$ be some EFG. We define a *subgame* $G_{ij}' = (\{i, j\}, \mathcal{A}, H, Z, A, P', (u_{ij}', u_{ji}'), \mathcal{I}, c, \pi_c)$ as a structurally identical game to $G$ between $i$ and $j$ with player function $P'$ and utility functions $(u_{ij}', u_{ji}')$, then

$$P'(h) \doteq \begin{cases} P(h) & \text{if } P(h) \in \{i, j\} \\ c & \text{o.w.} \end{cases}$$

and let $\pi_c'$ be the strategy of the chance player in $G_{ij}'$. We put no restrictions on $\pi_c'$.

Note that $u_{ij}', u_{ji}'$ are not necessarily defined in the above definition. They may take any values. We merely want the subgame $G_{ij}'$ to share the structure of $G$. In $G_{ij}'$, $i$ and $j$'s utility only depends on their strategies $\pi_i, \pi_j$ and chance's actions $\pi_c'$:

$$u_{ij}'(\pi_i, \pi_j) \doteq \mathbb{E}_{z \sim (\pi_i, \pi_j, \pi_j)} \left[ u_{ij}'(z) \right]$$
$$= \sum_{z \in Z} p_i(z, \pi_i) p_j(z, \pi_j) p_c(z, \pi_c') u_{ij}'(z).$$

What should $\pi_c'$ be defined as? This turns out to not matter very much. Given any subgame $G_{ij}'$ with chance strategy $\pi_c'$ and utility functions $u_{ij}', u_{ji}'$, for any $G_{ij}''$ with chance strategy $\pi_c''$, we can find $u_{ij}'', u_{ji}''$ so that the utility of players between the two games will always be equal for any strategy profile $(\pi_i, \pi_j)$.

**Definition E.5.** We say $\pi_c$ is *fully mixed* if $\pi_c'(a, h) > 0 \ \forall h \in \{h \in H \mid P(h) = c\}, a \in A(h)$

**Proposition E.6.** *Let $G$ be an EFG and $G_{ij}'$ a subgame between $i$ and $j$ with utility functions $u_{ij}', u_{ji}'$ and fully mixed chance strategy $\pi_c'$. Given $G_{ij}''$, a subgame between $i$ and $j$ with fully mixed chance strategy $\pi_c''$, we may find $u_{ij}'', u_{ji}''$ such that $\forall \pi_i, \pi_j$ we have $u_{ij}'(\pi_i, \pi_j) = u_{ij}''(\pi_i, \pi_j)$ and $u_{ji}'(\pi_i, \pi_j) = u_{ji}''(\pi_i, \pi_j)$.*

*Proof.* Note that $p_c(z, \pi'_c), p_c(z, \pi''_c) \neq 0$. Then $\forall z \in Z$, define $u''_{ij}(z) = \frac{p_c(z, \pi'_c)}{p_c(z, \pi''_c)} u'_{ij}(z)$ and $u''_{ji}(z) = \frac{p_c(z, \pi'_c)}{p_c(z, \pi''_c)} u'_{ji}(z)$. Then

$$\begin{aligned}
u''_{ij}(\pi_i, \pi_j) &= \sum_{z \in Z} p_i(z, \pi_i) p_j(z, \pi_j) p_c(z, \pi''_c) u''_{ij}(z) \\
&= \sum_{z \in Z} p_i(z, \pi_i) p_j(z, \pi_j) p_c(z, \pi''_c) \frac{p_c(z, \pi'_c)}{p_c(z, \pi''_c)} u'_{ij}(z) \\
&= \sum_{z \in Z} p_i(z, \pi_i) p_j(z, \pi_j) p_c(z, \pi'_c) u'_{ij}(z) \\
&= u'_{ij}(\pi_i, \pi_j).
\end{aligned}$$

$\square$

For the remainder of this work, when defining a subgame between $i$ and $j$ given an EFG $G$, we define the subgame chance player's strategy to be equal to $\pi_c$ in $G$ at information sets $I$ where $P(I) = c$ in $G$ and uniform randomly otherwise.

Having defined subgames, we may now define our representation of extensive-form polymatrix games.

**Definition E.7** (Poly-EFG). A poly-EFG $(N, E, \mathcal{G})$ is defined by a graph with nodes $N$, one for each player, a set of edges $E$ and a set of games $\mathcal{G} = \{G_{ij} \mid \forall (i,j) \in E\}$ where $G_{ij} \in \mathcal{G}$ is a two player EFG between $i$ and $j$ and all $G_{ij} \in \mathcal{G}$ are a subgame between $i$ and $j$ and all subgames are defined with respect to some EFG $G$.

Let $G_{ij} \in \mathcal{G}$ denote the subgame between $i$ and $j$. We use subscript $ij$ (e.g. $Z_{ij}$) to refer to parts of $G_{ij}$.

Since each subgame of the poly-EFG is the subgame of the same $G$, the space of pure, mixed and behavior strategies is the same for each subgame for any player. A player chooses a strategy (whether pure, mixed or behavior) and plays this strategy in each subgame. A player's global utility is the sum of their utility in each of their subgames. We have

$$u_i(\pi) = \sum_{(i,j) \in E_i} u_{ij}(\pi_i, \pi_j),$$

where $E_i \subseteq E$ is the set of edges connected to $i$ in the graph and $u_{ij}$ is the utility function of $i$ in subgame $G_{ij}$.

### E.3 Constant-Sum and Subgame Stable Poly-EFGs

Here, we give definitions of constant-sum and subgame stability for poly-EFGs. These definitions are largely identical to their normal-form counterparts, we merely provide them here for completeness.

Poly-EFGs are constant-sum if, for each subgame, the utilities at the terminals add up to a constant.

**Definition E.8** (Constant-sum). We say a poly-EFG $G$ is constant-sum if $\forall G_{ij} \in \mathcal{G}, z \in Z_{ij}$, $u_{ij}(z) + u_{ji}(z) = c_{ij}$ where $Z_{ij}$ is the set of terminal histories of $\forall G_{ij}$ and $c_{ij}$ is a constant.

We may also define approximate poly-EFGs in the same way as in normal-form games.

**Definition E.9** ($\delta$-constant sum poly-EFG). An EFG $G$ is $\delta$-constant sum poly-EFG ($\delta$-CSP) if there exists a constant-sum poly-EFG $\check{G}$ with global utility function $\check{u}$ such that $\forall i \in N, \pi \in \Pi$, $|u_i(\pi) - \check{u}_i(\pi)| \leq \delta$. We denote the set of such CSP games as $\mathrm{CSP}_\delta(G)$.

Finally, we define subgame stability for poly-EFGs. Our definitions are near-identical from the normal-form definitions.

**Definition E.10** (Subgame stable profile). Let $G$ be a poly-EFG. We say a strategy profile $\pi$ is $\gamma$-subgame stable if $\forall (i,j) \in E$, we have $(\pi_i, \pi_j)$ is a $\gamma$-Nash of $G_{ij}$.

**Definition E.11** (Subgame stable game). Let $G$ be a poly-EFG. We say $G$ is $(\epsilon, \gamma)$-subgame stable if for *any* $\epsilon$-Nash equilibrium $\pi$ of $G$, $\pi$ is $\gamma$-subgame stable.

### E.4 Theoretical Results For Poly-EFGs

Our theoretical results from Sections 4 and 5 continue to hold in poly-EFGs. The idea is to use the induced normal-form of a poly-EFG. We assume perfect recall.

First, we will characterize what self-play will produce in EFGs. These are called *marginal behavior strategies*.

**Definition E.12** (Marginal behavior strategy). Given some mediated equilibrium $(\mu, (\Phi_i)_{i=1}^N)$, let $\pi_i^\mu$ be the *marginal behavior strategy* for $i$ where $\pi_i^\mu(a, I)$ is defined arbitrarily if $\sum_{\rho_i' \in P_i(I)} s_i^\mu(\rho_i') = 0$ and otherwise

$$\pi_i^\mu(a, I) \doteq \frac{\sum_{\rho_i \in P_i(a,I)} s_i^\mu(\rho_i)}{\sum_{\rho_i' \in P_i(I)} s_i^\mu(\rho_i')} \quad \forall I \in \mathcal{I}_i, a \in A(I),$$

where $s_i^\mu(\rho_i) \doteq \sum_{\rho_{-i} \in P_{-i}} \mu(\rho_i, \rho_{-i})$.

**Definition E.13** (Marginal behavior strategy profile). Given some mediated equilibrium $(\mu, (\Phi_i)_{i=1}^N)$, let $\pi^\mu$ be a *marginal behavior strategy profile*, where $\pi_i^\mu$ is a marginal behavior strategy $\forall i \in N$.

**Definition E.14** (Induced normal-form polymatrix game). Given a poly-EFG $G = (N, E, \mathcal{G})$, the *induced normal-form polymatrix game* is a polymatrix game $G' = (N, E, P, u')$ such that $P_i$ is equal to $i$'s set of pure strategies in each $G_{ij}$ and $u'_{ij}(\rho_i, \rho_j) = u_{ij}(\rho_i, \rho_j) = \sum_{z \in Z} p_i(z, \rho_i) p_j(z, \rho_j) p_c(z, \pi_c') u_{ij}(z)$ where $u_{ij}$ is the utility function of $i$ in $G_{ij}$.

In games of perfect recall, every behavior strategy $\pi_i$ has an equivalent mixed strategy $s_i^{\pi_i}$ (by Kuhn's Theorem), which means for any perfect recall EFG $G$, we can use the poly-EFG representation instead of turning $G$ into a normal-form game then using a normal-form polymatrix game to get the same vulnerability bounds on $G$. Given $\pi$, let $s^\pi$ be a profile of equivalent mixed strategies.

From Kuhn's Theorem and an assumption that each $G_{ij} \in \mathcal{G}$ has perfect recall, we derive two immediate corollaries.

**Corollary E.15.** *If $\pi$ is $\gamma$ subgame stable for a poly-EFG $\check{G}$ where each subgame has perfect recall, then $s^\pi$ is $\gamma$ subgame stable in the induced normal-form polymatrix game of $\check{G}$.*

**Corollary E.16.** *For EFG of perfect recall $G$, if $G$ is $\delta$-CSP-EFG then the induced normal form of $G$ is $\delta$-CSP.*

An extension of Theorem 4.9 holds for poly-EFGs. Let $\Pi^\mu$ be the set of marginal behavior strategy profiles for any CCE of $G$ and $\Pi_i^\mu$ be the set of marginal behavior strategies for $i$.

**Proposition E.17.** *If an EFG $G$ is $\delta$-CSP and $\exists \check{G} \in \mathrm{CSP}_\delta(G)$ that is $(2n\delta, \gamma)$-subgame stable and $\mu$ is a CCE [5] of $G$, then*

$$\mathrm{Vul}_i\left(\pi^\mu, \Pi_{-i}\right) \leq |E_i|\gamma + 2\delta \leq (n-1)\gamma + 2\delta,$$

*Proof.* Transform $\check{G}$ into its induced normal-form polymatrix game $\check{G}'$. By Corollaries E.15 and E.16 the induced normal form of $G$ is $\delta$-CSP and $(2n\delta, \gamma)$-subgame stable. By perfect recall, we can convert $\pi^\mu$ to an equivalent mixed strategy profile $s^\mu$ and do likewise with any $\pi_{-i} \in \Pi_{-i}$. Then apply Theorem 4.9 using $s^\mu$, $S_{-i}$ and the induced normal-form polymatrix game of $\check{G}$ to bound vulnerability on $G$'s induced normal form, and hence $G$. $\qquad\square$

### E.5 Vulnerability Against Self-Taught Agents in EFGs

Next we show an analogue of Theorem 5.3 for extensive-form games. In Section 5 we defined $S(\mathcal{A})$ as the set of marginal strategy profiles for a no-regret learning algorithm $\mathcal{A}$. Many algorithms for EFGs will compute behavior strategies, so we use $\Pi(\mathcal{A}) \doteq \{\pi^\mu \mid (\mu, (\Phi_i)_{i \in N}) \in \mathcal{M}(\mathcal{A})\}$ as the set of marginal behavior strategy profiles of $\mathcal{M}(\mathcal{A})$ (recall that $\mathcal{M}(\mathcal{A})$ is the set of mediated equilibria reachable by learning algorithm $\mathcal{A}$. Then let $\Pi_i(\mathcal{A}) \doteq \{\pi_i \mid \pi \in \Pi(\mathcal{A})\}$ be the set of $i$'s marginal strategies from $\Pi(\mathcal{A})$. For example, if $\mathcal{A}$ is CFR, $\mathcal{M}(\mathcal{A})$ is the set of observably sequentially rational CFCCE Morrill et al. (2021b) and $\Pi(\mathcal{A})$ is the set of behavior strategy profiles computable by CFR.

---

[5]Note that "CCE" refers to a *normal-form* CCE (NFCCE) in the language of Farina et al. (2020).

Next, suppose each player $i$ learns with their own self-play algorithm $\mathcal{A}_i$. Let $\mathcal{A}_N \doteq (\mathcal{A}_1, ... \mathcal{A}_n)$ be the profile of learning algorithms; $\Pi^\times(\mathcal{A}_N) \doteq \bigtimes_{i \in N} \Pi_i(\mathcal{A}_i)$ be the set of all possible match-ups between strategies learned in self-play by those learning algorithms and $\Pi^\times_{-i}(\mathcal{A}_N) \doteq \bigtimes_{j \in -i} \Pi_j(\mathcal{A}_j)$ be the profiles of $-i$ amongst these match-ups.

**Definition E.18.** We say a game $G$ is $\delta$-*CSP in the neighborhood of* $\Pi' \subseteq \Pi$ if there exists a constant sum poly-EFG $\check{G}$ such that $\forall \pi \in \Pi'$ we have $|u_i(\pi) - \check{u}_i(\pi)| \leq \delta$. We denote the set of such CSP games as $\mathrm{CSP}_\delta(G, \Pi')$.

**Definition E.19.** We say a poly-EFG game $G$ is $\gamma$-*subgame stable in the neighborhood of* $\Pi'$ if $\forall \pi \in \Pi', \forall (i,j) \in E$ we have that $(\pi_i, \pi_j)$ is a $\gamma$-Nash of $G_{ij}$.

**Proposition E.20.** *If $G$ is $\delta$-CSP in the neighborhood of $\Pi^\times(\mathcal{A}_N)$ and $\exists \check{G} \in \mathrm{CSP}_\delta(G, \Pi^\times(\mathcal{A}_N))$ that is $\gamma$-subgame stable in the the neighborhood of $\Pi(\mathcal{A}_i)$, then for any $\Pi \in \Pi(\mathcal{A}_i)$*

$$\mathrm{Vul}_i\left(\pi, \Pi^\times_{-i}(\mathcal{A}_N)\right) \leq |E_i|\gamma + 2\delta \leq (n-1)\gamma + 2\delta.$$

The proof goes the same way as in Corollary E.17. Use the induced normal-form polymatrix game of $\check{G}$ and Theorem 5.3 to derive bounds for the induced normal form of $G$, which then apply to $G$.

### E.6 Leveraging the Poly-EFG Representation for Computing CSP Decompositions

The poly-EFG representation gives rise to more efficient algorithms for computing a poly-EFG decomposition. As a proof of concept, we show that in perfect information EFGs, we can write a linear program to compute the optimal polymatrix decomposition for an EFG that is exponentially smaller than LP 2 from Section D.2. Recall that $\underline{\delta}$ is the *minimum* value of $\delta$ such that a game is $\delta$-CSP. In an perfect information EFG, we can compute $\underline{\delta}$ with the following LP. The decision variables are $\underline{\delta}$, the values of $\check{u}_{ij}(z) \; \forall z \in Z, (i,j) \in E$ and $c_{ij} \; \forall (i,j) \in E$.

**LP 3**

$$\min \quad \underline{\delta}$$

$$\text{s.t.} \quad u_i(z) - \sum_{j \in -i} \check{u}_{ij}(z) \leq \underline{\delta} \quad \forall i \in N, z \in Z$$

$$u_i(z) - \sum_{j \in -i} \check{u}_{ij}(z) \geq -\underline{\delta} \quad \forall i \in N, z \in Z$$

$$\check{u}_{ij}(z) + \check{u}_{ji}(z) = c_{ij} \quad \forall i \neq j \in N, z \in Z$$

The trick is that in perfect information EFGs, each pure strategy profile leads to a single terminal. Hence, rather than having a constraint for each pure strategy profile, a constraint for each terminal will suffice. This leads to an exponential reduction in the number of constraints over LP 2.

## F  Details of SGDecompose

Next, we give the full details of SGDecompose. We give details here using our poly-EFG representation, since this is the representation we use in our experiments.

For each subgame $\check{G}_{ij}$ we store a single vector $\check{u}_{ij}$ where the entry $\check{u}_{ij}(z)$ gives the value of the utility for corresponding terminal $z$. We additionally store a constant $\check{c}_{ij}$ for each subgame. Player $i$ will use $\check{u}_{ij}$ when computing $\check{u}_{ij}(\pi_i, \pi_j)$:

$$\check{u}_{ij}(\pi_i, \pi_j) = \sum_{z \in Z} p_i(z, \pi_i) p_j(z, \pi_j) p_c(z, \pi_c) \check{u}_{ij}(z).$$

Whereas we compute $\check{u}_{ji}(\pi_i, \pi_j)$ as follows.

$$\check{u}_{ji}(\pi_i, \pi_j) = \sum_{z \in Z} p_i(z, \pi_i) p_j(z, \pi_j) p_c(z, \pi_c) \left(\check{c}_{ij} - \check{u}_{ij}(z)\right).$$

For simplicity, let $\check{u}$ denote the stacked vector of all $\check{u}_{ij}$ and $\check{c}_{ij}$. We additionally initialize $\check{G}$ as a fully connected graph.

The overall loss function which we minimize has two components: first, $\mathcal{L}^\delta$ is the error between the utility functions of $G$ and $\check{G}$; it is a proxy for $\delta$ in $\delta$-CSP.

$$\mathcal{L}^\delta(\pi; \check{u}, u) \doteq \sum_{i \in N} |\check{u}_i(\pi) - u_i(\pi)|$$

$$= \sum_{i \in N} \left| \left( \sum_{(i,j) \in E_i} \check{u}_{ij}(\pi_i, \pi_j) \right) - u_i(\pi) \right|.$$

The other component of the overall loss function, $\mathcal{L}^\gamma$, measures the subgame stability. First, we define $\mathcal{L}^\gamma_{ij}$, which only applies to a single subgame.

$$\mathcal{L}^\gamma_{ij}(\pi_{ij}, \pi^*_{ij}; \check{u}) \doteq \max\left(\check{u}_{ij}(\pi^*_i, \pi_j) - \check{u}_{ij}(\pi_{ij}), 0\right)$$
$$+ \max\left(\check{u}_{ji}(\pi_i, \pi^*_j) - \check{u}_{ji}(\pi_{ij}), 0\right).$$

Where $\pi_{ij} = (\pi_i, \pi_j)$ is a profile and $\pi^*_{ij} = (\pi^*_i, \pi^*_j)$ is a profile of deviations. Then

$$\mathcal{L}^\gamma(\pi, \pi^*; \check{u}) \doteq \sum_{(i,j) \in E} \mathcal{L}^\gamma_{ij}(\pi_{ij}, \pi^*_{ij}; \check{u}).$$

Algorithm 3 shows how to compute a subgame stable constant-sum polymatrix decomposition via SGD. As input, the algorithm receives a game $G$, a finite set of strategy profiles $\Pi'$, learning rate $\eta$, number of training epochs $T$, hyperparameter $\lambda \in [0, 1]$ and batch size $B$. First, we initialize $\Pi^\times$ as the set of all match-ups amongst strategies in $\Pi'$.

We then repeat the following steps for each epoch. First, we compute a best-response (for example, via sequence-form linear programming) to each strategy $\pi'_i$ in $\Pi'$ in each subgame; the full process is shown in Algorithm 4. After computing these best-responses for the current utility function of $\check{G}$, we try to fit $\check{u}$ to be nearly CSP in the neighborhood of $\Pi^\times$ and subgame stable in the neighborhood of $\Pi'$. Since $\Pi^\times$ is exponentially larger than $\Pi'$, we partition it into batches, then use batch gradient descent. We use the following batch loss function, which computes the average values of $\mathcal{L}^\delta$ and $\mathcal{L}^\gamma$ over the batches, then weights the losses with $\lambda$. Let $\Pi^b$ denote a batch of strategy profiles from $\Pi^\times$ with size $B$, the overall loss function is

$$\mathcal{L}(\Pi^b, \Pi', \Pi^*; \check{u}, u) \doteq \frac{\lambda}{B} \sum_{\pi \in \Pi^b} \mathcal{L}^\delta(\pi; \check{u}, u) + \frac{(1-\lambda)}{|\Pi'|} \sum_{\pi \in \Pi'} \sum_{\pi^* \in \Pi^*} \mathcal{L}^\gamma(\pi, \pi^*; \check{u}).$$

We take this loss and find its gradient with respect to $\check{u}$, then update $\check{u}$:

$$\check{u} \leftarrow \check{u} - \eta \cdot \nabla_{\check{u}} \mathcal{L}(\Pi^b, \Pi', \Pi^*; \check{u}, u).$$

We found that in practise the gradient can be quite large relative to $\check{u}$, which has the potential to destabilize optimization. This is alleviated by normalizing the gradient by its $l^2$ norm.

$$g \leftarrow \nabla_{\check{u}} \mathcal{L}(\Pi^b, \Pi', \Pi^*; \check{u}, u)$$
$$\check{u} \leftarrow \check{u} - \eta \cdot \frac{g}{\|g\|_2}$$

# G   Experiment Details

The codebase for our experiments is available at `https://github.com/RevanMacQueen/Self-Play-Polymatrix`.

## G.1   Leduc Poker

We ran SGDecompose 30 times, each time with its own set of 30 strategy profiles. These 900 strategy profiles are generated with CFR+ in self-play for 1000 iterations with random initializations. We

**Algorithm 3** SGDecompose with behavior strategies

**Input:** $G, \Pi', \eta, T, \lambda, B$
Initialize $\check{u}$ to all $0$
$\Pi^\times \leftarrow \bigtimes_{i \in N} \hat{\Pi}_i$
**for** $t \in 1...T$ **do**
    $\Pi^* \leftarrow \text{getBRs}(\check{G}, \Pi')$
    $\mathcal{B} \leftarrow$ partition of $\Pi^\times$ into batches of size $B$
    **for** $\Pi^b \in \mathcal{B}$ **do**
        $g \leftarrow \nabla_{\check{u}} \mathcal{L}(\Pi^b, \Pi', \Pi^*; \check{u}, u)$
        $\check{u} \leftarrow \check{u} - \eta \cdot \frac{g}{\|g\|_2}$
    **end for**
**end for**
{Lastly, output $\delta$ and $\gamma$}
$\delta \leftarrow \max_{\pi \in \Pi^\times} |u_i(\pi) - \check{u}_i(\pi)|$
$\gamma \leftarrow \max_{\pi \in \Pi'} \max_{i \neq j \in N \times N} (\check{u}_{ij}(BR_{ij}(\pi_j), \pi_j) - \check{u}_{ij}(\pi_i, \pi_j))$
**return** $\check{u}, \gamma, \delta$

---

**Algorithm 4** getBRs

**Input:** $\check{G}, \Pi'$
$\Pi_i^* \leftarrow \varnothing \; \forall i \in N$
**for** $i \neq j \in N \times N$ **do**
    **for** $\pi_j \in \Pi_j'$ **do**
        compute $\pi_{ij}^* \in \arg\max_{\pi_i' \in \Pi_i} \check{u}_{ij}(\pi_i', \pi_j)$
        $\Pi_i^* \leftarrow \Pi_i^* \cup \{\pi_{ij}^*\}$
    **end for**
**end for**
**return** $\bigtimes_{i \in N} \Pi_i^*$

---

randomly initialize CFR+ with regrets between 0 and 0.001 chosen uniformly at random, which are the default values in OpenSpiel.

Interestingly, we found that CFR+ converges to approximate Nash equilibria in Leduc poker, with a maximum value of $\epsilon$ equal to 0.013 after 1000 iterations. As we will show in Appendix G.2, CFR also empirically produces approximate Nash equilibria in Leduc poker.

Let $\Pi(\text{CFR+})_j$ denote the set of CFR+-learned strategy profiles for run $j$; and $\Pi^\times(\text{CFR+})_j$ denote the set of all match-ups between these 30 strategy profiles. Figure 5 shows diversity of $\Pi(\text{CFR+})_j$ for each of the 30 runs. We measure the diversity of each $\Pi(\text{CFR})_j$ by taking each pair of strategy profiles $\pi, \pi' \in \Pi(\text{CFR})_j$ and computing the total variation between these two probability distributions induced over the terminal histories of Leduc poker. We denote the maximum total variation for run $j$ as $TV_j$, where

$$TV_j \doteq \max_{\pi, \pi' \in \Pi(\text{CFR})_j} \frac{1}{2} \sum_{z \in Z} |p(z, \pi) - p(z, \pi')|.$$

$TV_j$ is constrained to be between 0 and 1, where 0 means the two distributions are the same and 1 means they are maximally different. Figure 5 shows the maximum total variation between any two of the strategy profiles used in each run.

We used the same parameters for each run of SGDecompose: $\lambda = 0.5$, $B = 30$, $T = 200$. We used a learning rate schedule where the learning rate $\eta$ begins at $2^{-6}$, then halves every 5 epochs until reaching $2^{-17}$ to encourage convergence. Our results are shown in Figure 6. We see that across the 30 runs of SGDecompose, Leduc poker is at most 0.021-CSP in the neighborhood of $\Pi^\times(\text{CFR+})_j$ and 0.009-subgame stable in the neighborhood of $\Pi(\text{CFR+})_j$. Figure 7 shows the maximum vulnerability with respect to the strategies in each of the runs compared to the bounds on vulnerability given by Proposition E.20. We compute the maximum vulnerability as

$$\text{Vul}_j \doteq \max_{i \in N} \max_{\pi \in \Pi(\text{CFR})_j} \text{Vul}_i \left( \pi, \Pi_{-i}^\times(\text{CFR})_j \right). \tag{10}$$

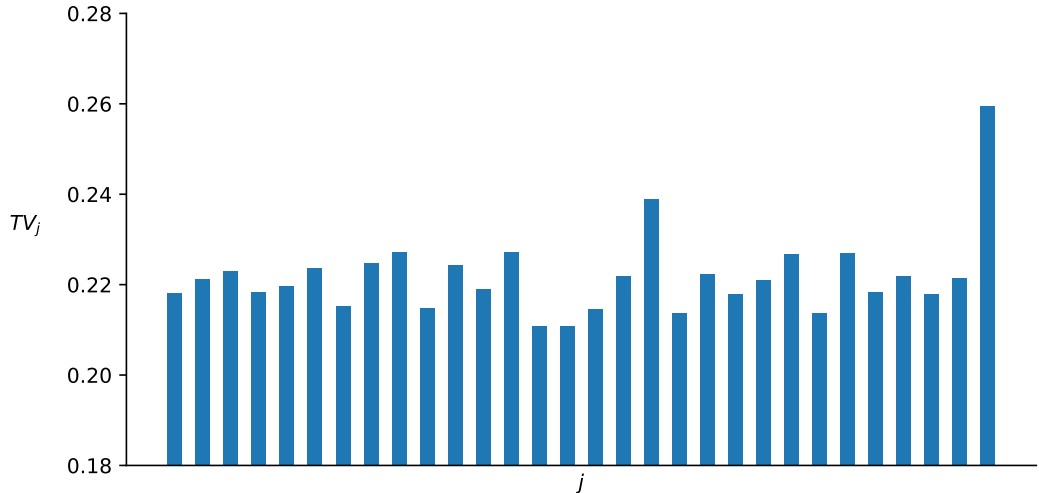

Figure 5: The maximum total variation for each $\Pi(\mathrm{CFR})_j$ used in different runs of SGDecompose in **Leduc Poker**. Different runs are shown on the x-axis, and the corresponding $TV_j$ for run $j$ is shown with the bars. A value of 0 indicates minimal diversity and 1 means maximal diversity. The minimum, mean, maximum and standard error across runs are 0.21, 0.22, 0.26 and 0.0016, respectively.

We see that the bounds are between 1.89 and 3.05 times the actual vulnerability, and are on average 2.51 times larger with a standard error of 0.049.

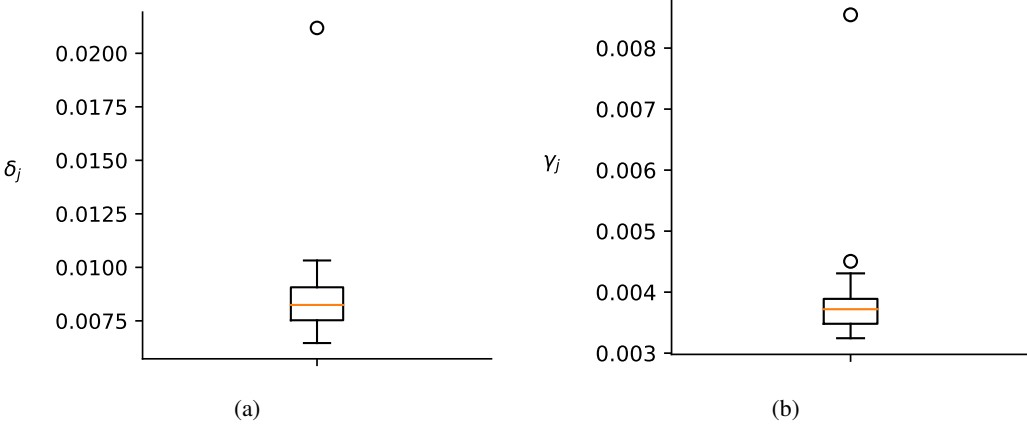

Figure 6: Boxplots showing the values of $\delta_j$ and $\gamma_j$ for each of the 30 runs of SGDecompose in **Leduc Poker**. Figure 6a shows the values of $\delta_j$, with the minimum, mean, maximum and standard error being 0.006, 0.009, 0.021 and 0.00046, respectively. Figure 6b shows the values of $\gamma_j$, with the minimum, mean, maximum and standard error being 0.003, 0.004, 0.009 and 0.00016, respectively.

### G.2 CFR Finds Approximate Nash in Leduc Poker

It was previously believed that CFR does not compute an $\epsilon$-Nash equilibrium on 3-player Leduc for any reasonable value of $\epsilon$. Previous work found that CFR computed a 0.130-Nash equilibrium after $10^8$ iterations Abou Risk & Szafron (2010). We saw in the previous section that CFR+ computes approximate Nash equilibria in Leduc poker—does this hold for CFR as well?

We ran 30 runs of CFR in self-play for 10,000 iterations and found that *all* of our strategies converged to an approximate Nash equilibrium with the maximum $\epsilon = 0.017$ after only $10^4$ iterations. Figure 8b

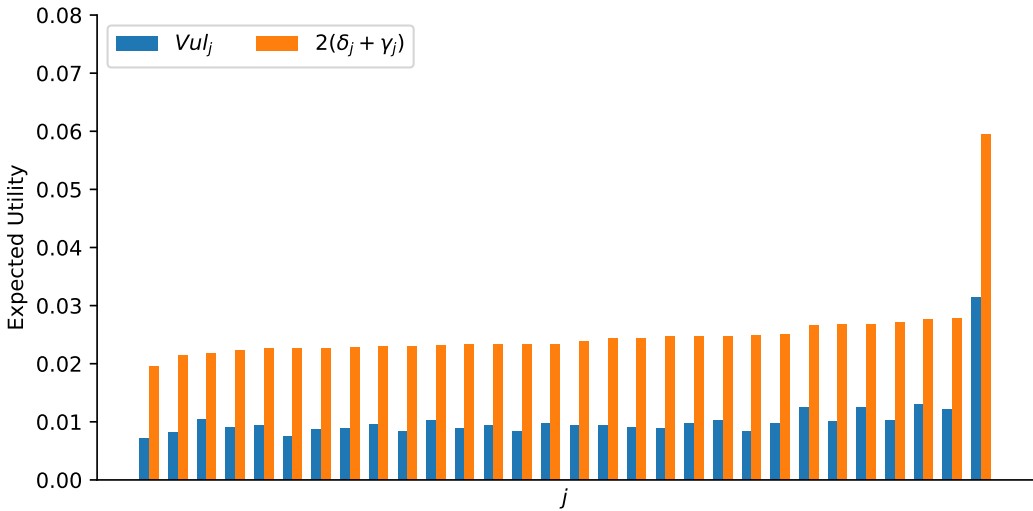

Figure 7: Bounds on vulnerability compared to true vulnerability in **Leduc Poker** for each run. Each of the 30 runs are shown on the x-axis. For each run $j$, we compute the bounds determined by Proposition E.20, which are $(n-1)\gamma_j + 2\delta_j = 2(\gamma_j + \delta_j)$. These value are shown in orange. The blue bars are the maximum vulnerability in each run, computed using (10). The ordering of bars in this plot matches the ordering of bars in Figure 5. The rightmost run had both the highest vulnerability and highest diversity.

shows the shows the maximum deviation incentive

$$\epsilon = \max_{\pi_i'} u_i(\pi_i', \pi_{-i}) - u_i(\pi)$$

for each of the CFR strategies $\pi$ computed by CFR in Leduc Poker. Each column is for one of the players and each point is one of the random seeds. We see the maximum value of $\epsilon$ after 10,000 iterations is 0.017. Figure 8a shows the maximum deviation incentive $\epsilon$ over 10,000 iterations. We average learning curves over 30 random seeds.

## H    Toy Hanabi

In games with low values of $\delta$ and $\gamma$, self-play will perform well against new opponents; however is the converse also true? Do games where self-play performs poorly against new opponents have high values of $\delta$ and $\gamma$? As mentioned earlier, self-play struggles to generalize to new agents in some games with specialized conventions (Hu et al., 2020). Hanabi is one such game (Bard et al., 2020). Hanabi is a cooperative game where players cannot see their own hands, but can see all other players hands; therefore players must give each other hints on how to act.

We show that a small version of the game of Hanabi is not close to the space of CSP games and self-play is quite vulnerable. We use Tiny Hanabi in the Openspiel framework (Lanctot et al., 2019) with our own payoffs, shown in Figure 9. Chance deals one of two hands, $A$ or $B$ with equal probability. Only player 1 may observe this hand and must signal to other players through their actions, $\sigma_1$ and $\sigma_2$, which hand chance has dealt. If both players 2 and 3 then correctly choose their actions to match chance's deal $((a, a)$ for $A$ or $(b, b)$ for $B)$ then all players get payoff equal to 1, otherwise all players get 0.

$\sigma_1$ and $\sigma_2$ can come to mean different things, $\sigma_1$ could signal to 2 and 3 to play $a$, or $b$. Self-play may learn either of these conventions. However, if a player trained in self-play encounters a set of players trained in an independent instance of self-play, they may not have compatible conventions.

This is indeed what happens when we train CFR on Tiny Hanabi in Figure 9. We trained 30 runs in self-play with different random initializations for 10,000 iterations. Some of these runs converged to

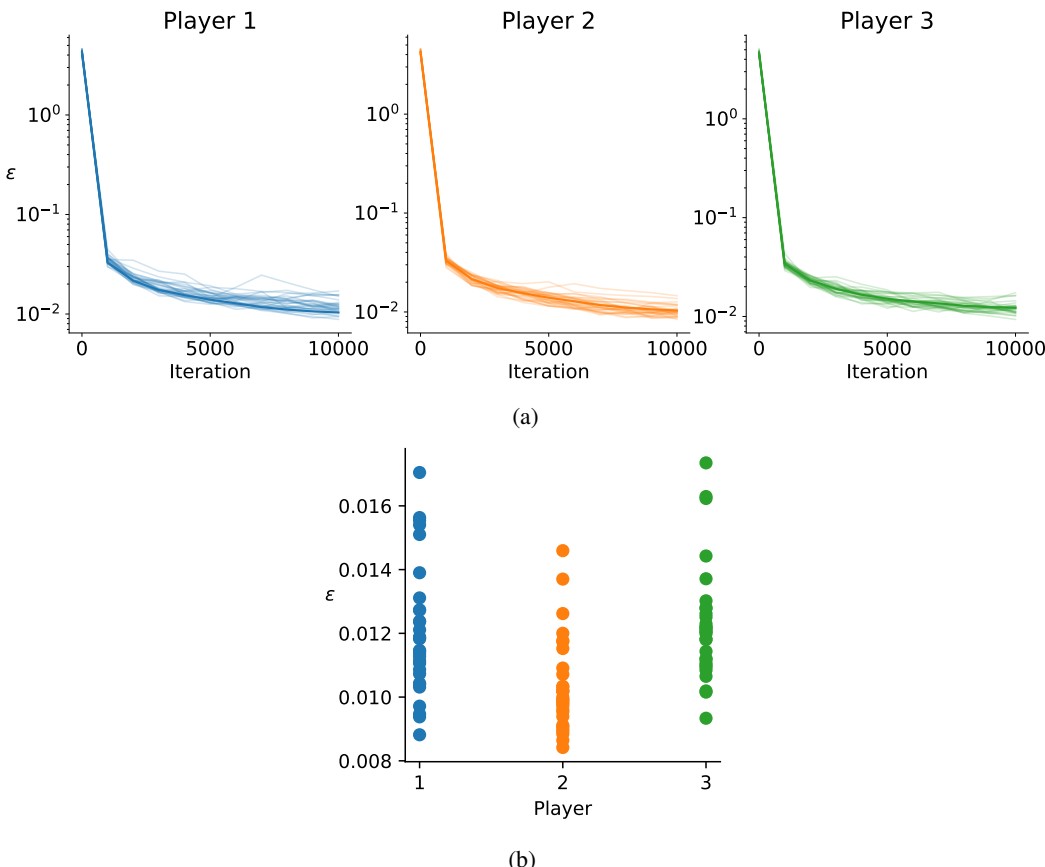

(a)

(b)

Figure 8: CFR empirically computes Nash Equilibria in Leduc Poker. (a) shows learning curves over iterations for each of the players. We measure $\epsilon$ by finding a best-response with sequence-form linear programming every 1000 iterations. We show each of the individual instances of CFR with different random initializations in light-coloured lines and the average across seeds in bold. (b) shows the distribution of $\epsilon$ at iteration 10,000.

each convention and when played against each other miscoordinated. We found

$$\max_{i \in N} \max_{\pi \in \Pi(\text{CFR})} \text{Vul}_i\left(\pi, \Pi^{\times}_{-i}(\text{CFR})\right) \approx 1,$$

as expected.

We decomposed Tiny Hanabi and found $\delta = 0.5$ and $\gamma \approx 0$, meaning the true vulnerability matched what our bounds predicted since $(n-1)\gamma + 2\delta \approx 1$. Why is $\gamma \approx 0$? We found that this was because our algorithm was setting the payoffs to equal to $0.50$ for all terminal histories, which is trivially polymatrix.

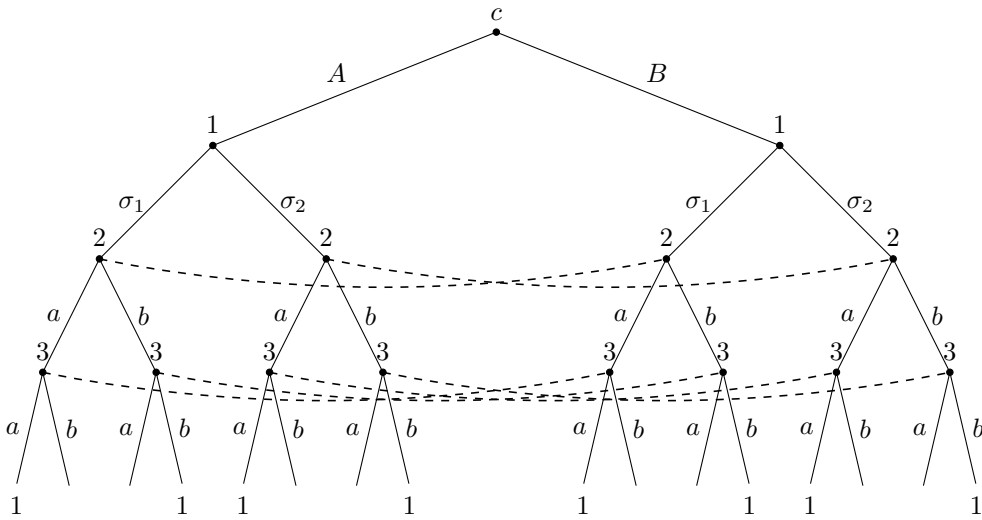

Figure 9: Tiny Hanabi. We omit payoffs of 0 at terminals.

