# OpenReview forum: "Guarantees for Self-Play in Multiplayer Games via Polymatrix Decomposability"
_NeurIPS.cc/2023/Conference — NeurIPS 2023 poster_

### Official Review · Reviewer_WEmp · 2023-06-26

**Soundness:** 3 good
**Presentation:** 2 fair
**Contribution:** 3 good
**Rating:** 6
**Confidence:** 2

**Summary:**

The paper studies theoretical performance guarantees for agents learned using self-play in multiplayer games. Self-play is a common approach of machine learning in multi-agent systems for generating unbounded quantities of training data, but agents trained using self-play may perform poorly against new agents whose behavior differ dramatically from those seen during training. Despite guarantees previously established for self-play agents in 2 player constant-sum games, these guarantees do not extend outside of two-player constant-sum games. To solve this problem, the authors identify a structural property of multiplayer, general-sum game and use it to establish guarantees on the performance of strategies learned via self-play against new opponents. They show that any game can be projected into the space of constant-sum polymatrix games, which enables performance guarantees for the strategies produced by a broad class of self-play algorithms. The findings are empirically demonstrated on Leduc poker.

**Strengths:**

- I find the studied topic which extends the theoretical guarantees of self-play beyond two-player (constant-sum) games to be important to the multi-agent learning community.
- The proposed method, by projecting a general game into the space of constant-sum polymatrix games, is novel and interesting.
- The theoretical results are verified on 3-player Leduc poker

**Weaknesses:**

The technical sections of the paper is slightly hard to follow for a non-expert in this field.

**Questions:**

- Can the authors pleaes explicitly list the assumptions/conditions needed for the theoretical results?
- Can the structural property identified in this paper extends to other application domains (outside of multi-agent self-play)?

**Limitations:**

I don't see negative societal impact.

---

> ### Author Rebuttal · Authors · 2023-08-10
>
> Thank you for your review.
>
> ### Response to Questions
> **Assumptions.** The main assumption we make is that self-play is performed by a no-regret learning algorithm. This means the average regret (the difference in utility between the chosen strategy and some hypothetical deviation) will get driven to 0 as the number of iterations increases. The set of deviations makes a difference on the type of resulting equilibrium, so we assume that the no-regret learning algorithms minimize external regret. This is a weak form of regret and almost every no-regret algorithm minimizes a stronger type of regret. We also assume that an agent learning via no-regret self-play will extract their strategy via marginalization in order to play it against new agents.
>
> **Extensions.** One of the fundamental problems in game theory is the equilibrium selection problem. Equilibria are fixed points where agents do not want to change their strategies; however if two agents found different equilibrium strategies (say, in self-play), then we do not generally know if the resulting joint selection of strategies is itself an equilibrium. This problem makes choosing a good strategy against new agents very hard.
>
> In two-player constant-sum games, all equilibria are exchangeable, which means that a joint selection of equilibrium strategies is also an equilibrium. This solves the equilibrium selection problem for this class of games. Subgame stable CSP games also solve the equilibrium selection problem in n-player games, since equilibria of the whole game also are equilibria of the subgames, which are two-player constant-sum and hence exchangeable. This means equilibria of the whole game are also exchangeable.

---

> > ### Comment · Reviewer_WEmp · 2023-08-11
> >
> > Thanks for taking the time to respond!
> > The clarification of the assumptions is useful to know, and thanks for commenting on the extensions!

---

### Official Review · Reviewer_wuxJ · 2023-07-05

**Soundness:** 4 excellent
**Presentation:** 4 excellent
**Contribution:** 4 excellent
**Rating:** 8
**Confidence:** 3

**Summary:**

This paper asks and answers the question: "In what games does self-play (with regret-minimizing algorithms) in multiplayer imperfect-information games perform well?"

The motivation behind the question is that such algorithms, such as multiplayer versions of CFR (which has theoretic guarantees in 2-player zero-sum games), have empirically performed well in some multiplayer games, despite no previous theory guaranteeing that they will do so. (The salient example is no-limit Texas Hold'em, in the work "Superhuman AI for multiplayer poker", Brown & Sandholm 2019.)

This paper answers the question primarily by looking at the concept of the coarse correlated equilibrium (CCE), since that is what the policies of the regret-minimizing players will converge to. The authors show that if games can be decomposed into a bunch of 2-player constant-sum games, called a constant-sum polymatrix (CSP) game, and the CSP game fulfills some additional properties, then we can bound the worst-case performance of a player's CCE strategy. They also show that even in games which can't be factored into polymatrix games (which is most games), we can project them into the space of CSP games, and still bound the worst-case performance of each player's CCE strategy.

They also refine the analysis further by examining worst-case performance only against other self-play strategies. Finally, they perform experiments on multiplayer Leduc poker, and show that the empirical worst-case performances are within the bounds predicted by their theory.

**Strengths:**

- I think this is an excellent paper. The question it asks and answers is fascinating, and one that I think was begging to be examined for several years now. Therefore, I judge this paper to be a significant contribution to science.
- The results of the paper are novel and nontrivial.
- The paper is well-written. The writing is clear and understandable. The introduction explains the background and motivates the problem well.


**Weaknesses:**

The paper mentions no-limit Texas hold'em as a motivating real-world positive example of self-play working in multiplayer games. I also suggest that the authors mention some real-world negative examples -- for example, FAIR's line of work on Diplomacy, particularly the self-play agents performing poorly against humans in "No-Press Diplomacy from Scratch", Bakhtin et al 2021 https://arxiv.org/pdf/2110.02924.pdf (the problem is the motivation of "Mastering the Game of No-Press Diplomacy via Human-Regularized Reinforcement Learning and Planning" Bakhtin et al 2022: "the resulting agent DORA does very well when playing with other copies of itself. However, DORA performs poorly in games with 6 human human-like agents.")

typo on line 336: "with ... be the set"

**Questions:**

It's not clear to me how important the caveat in Footnote 3 on page 8 (section 6) is. So CFR does not necessarily converge to a CCE. The experiments in Section 6 use CFR. Is my interpretation correct that you take the marginal strategies of the CFR iterates, and hand-wavingly say that we can assume they are marginal strategies of (a distribution converging to)
a CCE, even though they're not?

**Limitations:**

The authors adequately addressed the limitations.

---

> ### Author Rebuttal · Authors · 2023-08-10
>
> Thank you for your encouraging review.
>
> ### Response to Weaknesses
> We think this is an excellent suggestion for our work, and we will definitely mention this. Since submission, we conducted additional experiments on a toy version of Hanabi (another game where self-play is known to perform well in training but poorly against new agents) and found that this game was not well-approximated by a CSP game.
>
> ### Response to Questions
> The footnote was in order to clarify a detail about the implementation of CFR. The OpenSpiel CFR is indeed guaranteed to produce the marginal strategies of a CCE, since the average of the marginal strategies is equal to the marginal of the average strategy profile. However, if one wanted to extract the actual joint distribution across pure strategies, one would need to use a method like CFR-JR. We will clarify this detail.

---

> > ### Comment · Reviewer_wuxJ · 2023-08-10
> > **response**
> >
> > Thanks for taking the time to respond.
> >
> > Glad to hear that my suggestion is good.
> >
> > The Hanabi experiments sound useful.
> >
> > Re: response to questions -- I think I understand now. Is this a correct take?: The distribution of joint strategies produced by CFR iterates does converge to a CCE. However, the footnote is explaining that if you take each player's marginal/average strategy, the strategy profile that you get from combining all of those is not a CCE. But this is fine, because the whole point of the paper is to analyze those marginal strategies anyways.

---

> > > ### Author Response · Authors · 2023-08-11
> > >
> > > Yes that is correct!
> > >
> > > Just to add a couple details: CFR iterates are behavior strategies, whereas CCE are distributions over pure strategy profiles. This is why you need an extra step with CFR-JR to convert the behavior strategies to equivalent mixed strategies when you want to extract the empirical distribution of play. But you don't need to do this to extract the marginals in behavior strategy form.
> > >
> > > The marginals *could* be CCE themselves (if they are a Nash equilibrium) but this is not necessarily the case.

---

### Official Review · Reviewer_Wd36 · 2023-07-07

**Soundness:** 3 good
**Presentation:** 3 good
**Contribution:** 3 good
**Rating:** 7
**Confidence:** 3

**Summary:**

In multiplayer games, the authors derive bounds for the vulnerability of marginal strategies trained via no-regret self-play against other, uncorrelated agents independently trained via no-regret self-play. This is done by projecting games onto the space of constant-sum polymatrix (CSP) games, which can be decomposed into a set of 2-player constant-sum games between individual players. The closer this projection is, the tighter the bound on self-play's vulnerability to other self-play-trained agents is. This claim is validated by demonstrating that 3-payer Leduc can indeed be closely approximated as a CSP game and that the calculated vulnerability bounds are relatively close to the empirically measured vulnerability among marginal CFR strategies from many seeds.

**Strengths:**

- A bound on the vulnerability of marginal no-regret strategies to other uncorrelated no-regret strategies is novel and highly useful to the game theory/AI community.
- The paper is well written and concepts are clearly explained.

**Weaknesses:**

- I would have preferred to see a range of common tractable games examined to get an intuition on when and how often a game can be closely approximated as a CSP game (and thus provide a useful bound on vulnerability). Currently, experiments only include Leduc Poker.
- The analysis method is not immediately transferable to large games, and this is clearly stated as a limitation.


- Minor note: It's not explicitly stated until the very end of section 6 that the 3-player variant of Leduc Poker is used for experiments.

**Questions:**

Are there any heuristics one may be able to look for in larger games that might indicate when a game has a high chance of being approximately decomposable as a CSP game? If there is intuition on this to be gained from explicitly analyzing smaller games, it would be a great thing to discuss.

**Limitations:**

All limitations have been adequately addressed.

---

> ### Author Rebuttal · Authors · 2023-08-10
>
> Thank you for your review and interesting questions.
>
> ### Response to Weaknesses
> 1. We agree that it would be interesting to validate our approach on a larger suite of games. Since submission, we have conducted experiments on a toy Hanabi game and found that self-play performs poorly against new opponents and the toy Hanabi game is not well-approximated by a CSP game.
>
> 2. We think better algorithms for decomposing large games is a great direction for future research. Our analysis still applies if you can analytically show that a large game is subgame stable CSP. Since submission, we have also improved our algorithm which allows us to more efficiently decompose Leduc poker, which already has about 25,000 information sets.
>
> 3. We will add this clarification.
>
> ### Response to Questions
> We think that poker might be well-approximated by a CSP game because for most hands, all but two players fold relatively quickly, which means the game really looks like a set of two-player games. Bad Card shows this intuition by having the dominated strategy reduced game being CSP, but the overall game isn’t. One could also look for independence between the interactions between agents. For example, if an agent were to simultaneously play two games of chess, this overall “game” would be CSP.

---

> > ### Comment · Reviewer_Wd36 · 2023-08-15
> >
> > Thank you for your replies. Adding Hanabi is a great addition. That addresses the one meaningful weakness I had with this work.
> >
> > I also appreciate that you plan to clearer distinguish that, by 'self play', you mean 'no-regret self play' as hBPZ mentioned. This could lead to a small amount of initial confusion for some readers.

---

### Official Review · Reviewer_hBPZ · 2023-07-07

**Soundness:** 3 good
**Presentation:** 3 good
**Contribution:** 3 good
**Rating:** 6
**Confidence:** 4

**Summary:**

This paper explores the intriguing problem of why no-regret algorithms seem to approximate well in multiplayer games, a phenomenon that has been empirically demonstrated in multiagent poker. The authors identify a structural property in multi-player games that allows performance guarantees for strategies derived through self-play algorithms. They propose that multi-player games can be projected into a series of two-player constant-sum games, known as polymatrix games. The proximity of a game to this structure is hypothesized to diminish the effect of correlation issues on the removal of a mediator. The researchers take an algorithm-agnostic approach, which broadens the applicability of their analysis to a variety of game-theoretically inspired learning algorithms and MARL algorithms that converge to coarse correlated equilibria.


**Strengths:**

The paper's strength lies in its novel approach to an important problem. The authors introduce theory that all multiplayer games can be projected onto the space of polymatrix games, offering a promising direction for further research. If a high subgame stability game exists within this space, no-regret learning algorithms are predicted to converge to low-exploitability strategies. Additionally, the authors' algorithm-agnostic approach ensures that the analysis remains broadly applicable to a variety of game-theoretically inspired learning and MARL algorithms.


**Weaknesses:**

The paper's main weaknesses lie in its experimental section, which comes across as somewhat unclear. The section seems intended to demonstrate that CFR will always converge to a low-exploitable strategy due to the subgame stability of the game, but this is not clearly stated. If the goal is to show that no-regret algorithms will always converge, the experiment design should include many seeds with different hyperparameters and many different no-regret algorithms. Secondly, it's unclear how this theory can be practically applied. For instance, if a new game is introduced, can it be predicted ahead of time whether CFR or another no-regret algorithm will converge to a low-exploitability strategy? This potential application isn't clear and should be further highlighted if it is indeed feasible.


**Questions:**

How can the theory be applied practically? Can it be used to predict whether a no-regret algorithm will converge to a low-exploitability strategy in a new game?
Could the use of the term 'vulnerability' over 'exploitability' be explained or justified in this context?
What are the detailed parameters and specifics of the experiments conducted, such as the number of seeds and the range of no-regret algorithms tested?


**Limitations:**

There are certain limitations in the terminology used in the paper. The authors use 'vulnerability' instead of 'exploitability', although 'exploitability' is the term most widely used and recognized in this field. In addition, the term 'self-play' can lead to confusion, as it can generally refer to methods where RL agents play against themselves, which are not no-regret. The term 'no-regret' would be clearer. Lastly, at the end of page 9: “and if there exists a game with this set with high subgame stability” do you mean “and if there exists a game in this set with high subgame stability”?

---

> ### Author Rebuttal · Authors · 2023-08-10
>
> Thank you for your thoughtful review.
>
> ### Response to Questions
> **Practical applications.** This work can be used practically to show in which multiplayer games pre-computing a strategy via self-play is desirable. It absolutely can be used to predict whether the strategy of a no-regret algorithm will have low vulnerability. We know that for any two-player constant-sum games, strategies learned via no-regret self-play have performance guarantees. We see the practical use of our theory in much the same way: showing a particular game/application is subgame stable CSP will give similar theoretical guarantees.
>
> Showing a game has these properties can be done analytically by reasoning about the structure of a game. One could also sample strategies and empirically test whether a game is subgame stable CSP in the neighborhood of these strategies using our algorithm. This approach could quickly build intuition about a game before formally proving it is subgame stable CSP.
>
> We also think our work could be applied in mechanism design for multiplayer settings. A mechanism designer could be sure to design a game to be subgame stable CSP. Behavior of agents would likely be more predictable and stable.
>
> **Vulnerability vs Exploitability.** We chose to use the term “vulnerability” over “exploitability” in our work since “exploitability” already has a different meaning in the literature for n-player games. For example in this paper (https://www.ijcai.org/proceedings/2022/0484.pdf) exploitability is defined as the average incentive to deviate across players. This is a different quantity than vulnerability, which we define as the difference in utility between a strategy profile and the worst-case joint deviation by $-i$.
>
> **Experiment Details.** We tested a single no regret algorithm: vanilla CFR (the code is here: https://github.com/deepmind/open_spiel/blob/master/open_spiel/python/algorithms/cfr.py). We used CFR with simultaneous updates. The only modification we made was to allow random initialization of CFR’s initial strategy, which is uniformly random by default. We used 30 random seeds. We chose CFR since it is a widely used algorithm and efficient for large games.
>
> ### Response to Limitations
> We will clarify the distinction between the usual RL self-play and no-regret self-play.
>
> We do mean the latter sentence. Thank you for catching that.

---

> > ### Comment · Reviewer_hBPZ · 2023-08-14
> > **Response to Authors**
> >
> > Thanks for answering my questions. After reading the rebuttal and other reviews I choose to keep my score at a 6.

---

### Decision · Program_Chairs · 2023-09-21

**Decision:**

Accept (poster)

**Comment:**

The paper studies the structural properties of multiplayer games where self-play is guaranteed to perform well against opponents. The paper contains new ideas, is well-motivated, and well-written. It reaches a consensus that this is a good paper that should be accepted. Please make sure all the comments are addressed in preparing the final version.